# Global hotspots of salt marsh change and carbon emissions

Anthony D. Campbell[1,2,3 ✉], Lola Fatoyinbo[1], Liza Goldberg[1,4] & David Lagomasino[5]

Salt marshes provide ecosystem services such as carbon sequestration[1], coastal protection[2], sea-level-rise (SLR) adaptation[3] and recreation[4]. SLR[5], storm events[6], drainage[7] and mangrove encroachment[8] are known drivers of salt marsh loss. However, the global magnitude and location of changes in salt marsh extent remains uncertain. Here we conduct a global and systematic change analysis of Landsat satellite imagery from the years 2000–2019 to quantify the loss, gain and recovery of salt marsh ecosystems and then estimate the impact of these changes on blue carbon stocks. We show a net salt marsh loss globally, equivalent to an area double the size of Singapore (719 km²), with a loss rate of 0.28% year⁻¹ from 2000 to 2019. Net global losses resulted in 16.3 (0.4–33.2, 90% confidence interval) Tg $CO_2$e year⁻¹ emissions from 2000 to 2019 and a 0.045 (−0.14–0.115) Tg $CO_2$e year⁻¹ reduction of carbon burial. Russia and the USA accounted for 64% of salt marsh losses, driven by hurricanes and coastal erosion. Our findings highlight the vulnerability of salt marsh systems to climatic changes such as SLR and intensification of storms and cyclones.

Salt marshes provide essential ecosystem services such as carbon sequestration[1], coastal protection[2], SLR adaptation[3] and recreation[4]. Salt marshes, mangroves and seagrass are commonly called blue carbon ecosystems—coastal wetlands that store and sequester large amounts of carbon[1]. Globally, salt marshes occur across all continents, except Antarctica, in low-energy tidal environments. About 40% of the mapped salt marsh extent is found in North America and about 25% in Australia[9]. Globally, coastal wetland loss rates increased for much of the twentieth century before declining in the 1990s[10]. Drivers of salt marsh loss include drainage[7], eutrophication[11], sediment availability[12] and SLR[5]. SLR-driven marine transgression can also cause gains[13,14], an important process to offset loss globally[15]. Recent studies have addressed the difficulty of global mapping and change analysis in intertidal and subtidal systems with cloud computing including mangroves and tidal flats[16–19], but salt marsh monitoring activities are limited to national efforts[20] or included in generalized global estimates of coastal wetland change[21]. Moreover, the latest blue carbon accounting of stocks and fluxes still relies on dated estimates of salt marsh change (1–2% year⁻¹) derived from limited in situ analyses of several estuaries and century-long time periods[22]. Here we create the first consistent spatial and temporal estimates of contemporary salt marsh change from 2000 to 2019.

We analyse the global distribution of salt marsh change rates, including loss, gain and recovery. We also assess the impact of these changes on salt marsh carbon stocks worldwide from the year 2000 onward. We constrained our analysis with the most comprehensive global salt marsh map available, based on a compilation of national and regional datasets[9]. We processed all Landsat 5, 7 and 8 imagery with Google Earth Engine within 1.8 km of the known extent[9] by implementing a normalized difference vegetation index (NDVI)-based anomaly analysis[17,18], comparing a reference period (1984–1999) to change in four 5-year epochs (2000–2019). We further conducted a rigorous accuracy assessment of our analyses with 12,600 validation points split evenly by epoch and used to calculate confidence intervals and threshold sensitivity. A panel regression analysis was also conducted by watershed for conterminous United States (excluding Alaska, Hawaii and Puerto Rico), subsequently referred to as the USA, to understand change drivers, including storm events, urbanization, change surrounding the salt marsh and local sea-level change (LSLC), defined as the 5-year local trend in sea level. Our global salt marsh change data are openly available.

From 2000 to 2019, there was a global net salt marsh loss of 1,452.84 km² (733.1–2,172.07 km²; Fig. 1). This net salt marsh loss is equivalent to a quarter of net mangrove losses (5,807.1 km²) from 1996 to 2016 (ref. [23]), in a global study of mangrove carbon emissions with areal change calculated from Earth observation[24]. Between 2005 and 2009, North America experienced the largest net loss of any region in a single epoch (282.6 km²). Here we found that watersheds affected by higher-category hurricanes lost more salt marsh. This highlights the climate dependence of these systems and expected increases in losses from climate change owing to increases in storm intensity and frequency. High uncertainty and continued net losses of salt marsh also highlight the need for continued global and local mapping and monitoring efforts at appropriate spatial and temporal resolutions to enable management, protection and restoration of these ecosystems.

## Hotspots of salt marsh change

Globally, an area of salt marsh approximately the size of two soccer fields (14,280 m²) was lost hourly from 2000 to 2019, totalling 2,733.33 ± 355.06 km². This loss was offset by 1,279.84 ± 255.34 km²

[1]Biospheric Sciences Laboratory, National Aeronautics and Space Administration (NASA) Goddard Space Flight Center, Greenbelt, MD, USA. [2]NASA Postdoctoral Program, Oak Ridge Associated Universities, Oak Ridge, TN, USA. [3]GESTAR II, University of Maryland, Baltimore County, Baltimore, MD, USA. [4]Earth System Science Interdisciplinary Center, University of Maryland, College Park, MD, USA. [5]Integrated Coastal Programs, East Carolina University, Wanchese, NC, USA. ✉e-mail: anthony.d.campbell@nasa.gov

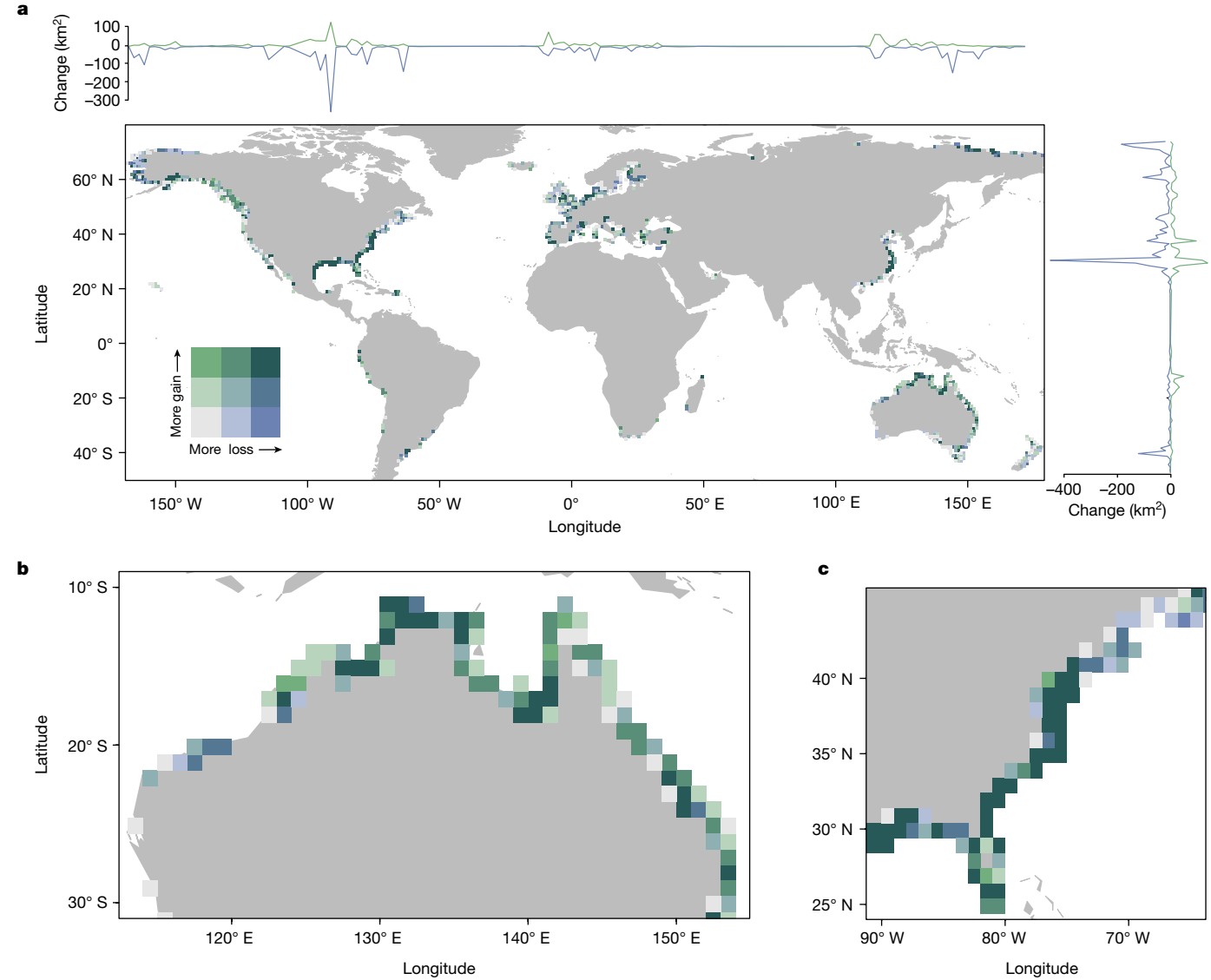

**Fig. 1 | Maps of global salt marsh change from 2000 to 2019. a**, Global salt marsh loss and gain map for 2000–2019 visualized with a three-quantile bivariate colour ramp. Line plots of gain and loss (km²) by longitude and latitude on the $x$ and $y$ axes, respectively. **b**, Salt marsh gain and loss in Northern Australia. **c**, Salt marsh gains and losses across the Atlantic coast and the Gulf of Mexico. Maps generated using R package ggplot2 and cowplot.

and 110.56 ± 20.05 km² of gain and recovery, respectively, for a net loss of 1,452.84 (733.1–2,172.07) km² (Fig. 2 and Supplementary Table 1). For comparison, from 1996 to 2016, there were 8,050.4 km² and 2,243.3 km² of mangrove loss and gain, respectively[23]. Our estimate of the global loss rate of salt marsh was 0.28% year⁻¹, a substantial reduction compared with loss rates of 1–2% year⁻¹ used in previous carbon emission estimates[22]. Net loss was slightly higher than mangrove net loss from 1996 to 2016 (0.2%)[23].

Salt marsh losses were most prominent in Russia and the USA, which accounted for 64% of the total global salt marsh loss (Extended Data Table 1). The epoch with the greatest global loss rate was 2015–2019, when salt marsh extent decreased at a rate of 0.33% year⁻¹, mainly owing to the large losses in Russia and the USA. In fact, the magnitude of marsh losses in Russia and North America from 2015 to 2019 were similar, despite the extent of Russian salt marsh being less than half that of North America. Extreme erosion rates (up to 20 m year⁻¹)[25], field survey methods, a starting survey year of 1973 and limited satellite data availability[9] were probably causes of the high loss rates of Russia. For the USA, salt marsh changed at a rate of −0.35% year⁻¹ from 2005 to 2009, which closely agrees with the loss rate for 2004–2009 from national

monitoring programmes (−0.46% year⁻¹)[20]. Epochs of elevated loss or gain were common globally. South America experienced elevated losses from 2000 to 2004 (Extended Data Table 1 and Supplementary Table 1). Oceania and Africa/Middle East were the only two regions in which marsh gains exceeded losses (Extended Data Table 1).

## Marsh recovery

Recovery from disturbances and landward migration are two critical components that influence the persistence of salt marshes but are poorly understood at both regional and global scales. Globally, 4.7% of all salt marsh losses had recovered by 2019, with most of the recovery occurring in areas lost between 2005 and 2009. These 2005–2009 losses coincide with extreme weather events such as hurricanes Rita, Wilma and Katrina in 2005, which greatly affected the Gulf Coast of the United States and resulted in a conversion of 562 km² of land to water in Louisiana[26]. The 16.5% recovery rate for losses occurring from 2005 to 2009 in the Gulf of Mexico region provides further evidence that storm events had a higher recovery rate than other loss drivers. Recovery increased in each subsequent epoch, except for the losses

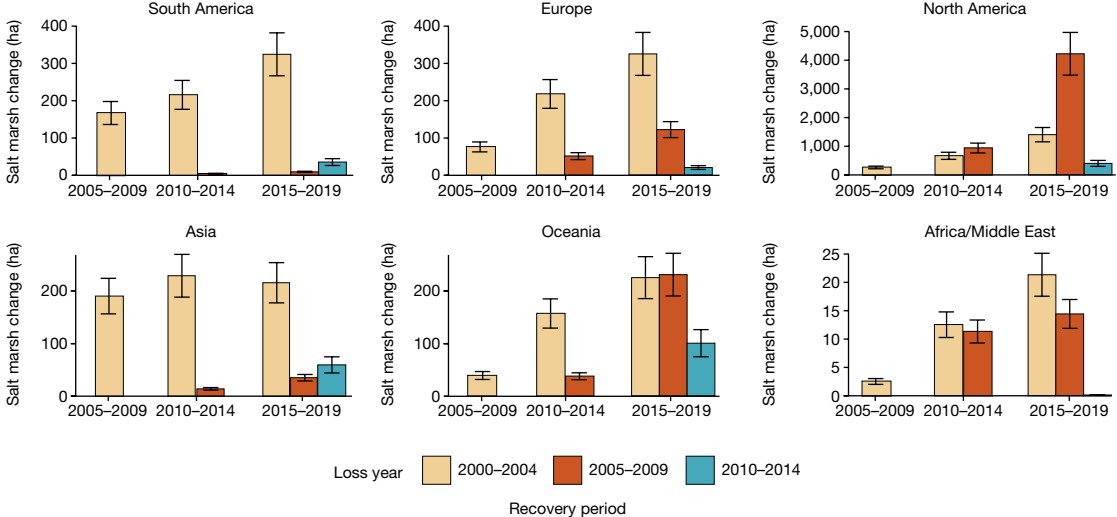

**Fig. 2 | Regional salt marsh recovery.** Salt marsh recovery for each study region by epoch in which the recovery occurred. Colour denotes the year in which a loss occurred. Error bars represent the standard error of the recovery area.

of Asia from 2000 to 2004 (Fig. 2). North American salt marsh comprised approximately 44% of the total salt marsh extent but 71% of all recovery. Still, salt marshes in the region were far from returning to the pre-epoch extents.

The location of loss and recovery times can provide essential insight into the process, type and amount of greenhouse gas emissions from blue carbon systems[27]. In addition to natural regrowth, the recovery maps probably also captured restoration sites. In North America, storm events in 2005–2009 resulted in high losses (Extended Data Table 1) and subsequent high recovery rate owing to a combination of restoration and natural revegetation. In 2007, 2 years after Hurricane Katrina made landfall, the Louisiana legislature responded by commissioning the first Coastal Master Plan, which resulted in coastal restoration projects such as marsh creation, sediment pipelines, shoreline restoration and oyster reef restoration[28]. The area of direct restoration is unclear given the potential indirect benefit of oyster reef restoration and sediment pipelines. Our results were able to quantify the long-term legacy of these recovery processes, which are critical for understanding salt marsh resilience.

## Salt marsh change drivers

We analysed salt marsh change within the USA in relation to LSLC, urbanization, change surrounding the salt marsh and hurricane landfall and intensity. We found that urbanization was not a detectable twenty-first century loss driver, suggesting that protections for salt marshes effectively limited conversion from drainage, and indirect effects related to urbanization such as increased nutrients and changes to the sediment supply were not considerable drivers of change. Similarly, findings in mangrove ecosystems showed that settlement accounted for only minimal recent losses[17]. The largest increase in loss was related to hurricane landfall and intensity, which increased salt marsh losses but had no notable effect on salt marsh gains.

In both panel regression models, loss and gain in the 100 m surrounding salt marshes were important predictors of salt marsh change (Supplementary Table 2). Losses surrounding the marsh were probably because of edge erosion and wetter tidal flats. Gain anomalies, increases in NDVI, in the 100 m surrounding the salt marsh, resulted in more changes within the salt marsh in terms of both losses as well as gains. For salt marsh loss, the observed relationship with vegetative greening (gain in NDVI) adjacent to a salt marsh could correspond to accretionary coasts (Fig. 3a–d). Higher LSLC was substantially related to reduced salt marsh gains (Supplementary Table 2).

The significance of change near the salt marsh in the watershed-scale panel analysis demonstrates the importance of gradual local change that our anomaly analysis observes. For example, SLR probably caused extensive losses within this section of Maryland's Eastern Shore (Fig. 3) supported by the nearby Ocean City tidal gauge with a long-term SLR trend of 6.05 ± 0.73 mm year$^{-1}$ (ref. [29]). Gradual loss is also evident, such as erosion along the barrier island (Fig. 3d). In the panel analysis, loss was not substantially affected by LSLC. As inundation increases in the region, loss anomalies surrounding the marsh increase, therefore these losses surrounding the marsh better reflect the impact of SLR than the short-term trends of LSLC.

Marsh erosion is linearly related to water body size[30] and, although not directly included in the analysis, we expect erosion rates to relate to losses of vegetation surrounding the marsh. Drainage and direct anthropogenic conversion were relatively limited in the USA. By contrast, on a global scale, salt marsh trends were complicated by anthropogenic change, which we believe is underrepresented in this analysis owing to the limitations of the baseline salt marsh extent dataset[9]. For example, a recent study of salt marsh change in China demonstrated a net loss of 359.27 km$^2$ from 1985 to 2019 but only 22.02 km$^2$ of loss from 2000 to 2019 (ref. [31]). Despite our analysis including only approximately half the salt marsh extent in China for 2000 (514 km$^2$ compared with 1,176 km$^2$), we found a similar small net loss rate of 0.006 (−0.45 to 0.47)% year$^{-1}$ and Chen et al. found a loss rate of 0.0009% year$^{-1}$ (ref. [31]). Similarly, a regional analysis of European salt marsh change found a net increase of 127.5 km$^2$ of salt marsh from 1986 to 2010 (ref. [32]), whereas our analysis found a 135.9 (38.7–235.8) km$^2$ loss of salt marsh extent from 2000 to 2019. Despite these differences in results between our maps and more localized studies, our results allow, for the first time, to evaluate global patterns of salt marsh change with a consistent dataset, reproducible methodology and rigorous uncertainty analysis.

## Uncertainty and future analysis

Our work focused on improving salt marsh change estimates only. However, overlap in blue carbon ecosystem extent can introduce some uncertainty. Globally, there is overlap between the global extent of mangroves[24] and the mapped extent of salt marshes[9]. Most of this overlap, 80%, occurs within Australia (1,590 km$^2$). To complicate matters, the overlap between the two ecosystems is a source of double-counting in existing blue carbon budgets, and overlap should be accounted for in uncertainty estimates. Mangrove encroachment is probably an

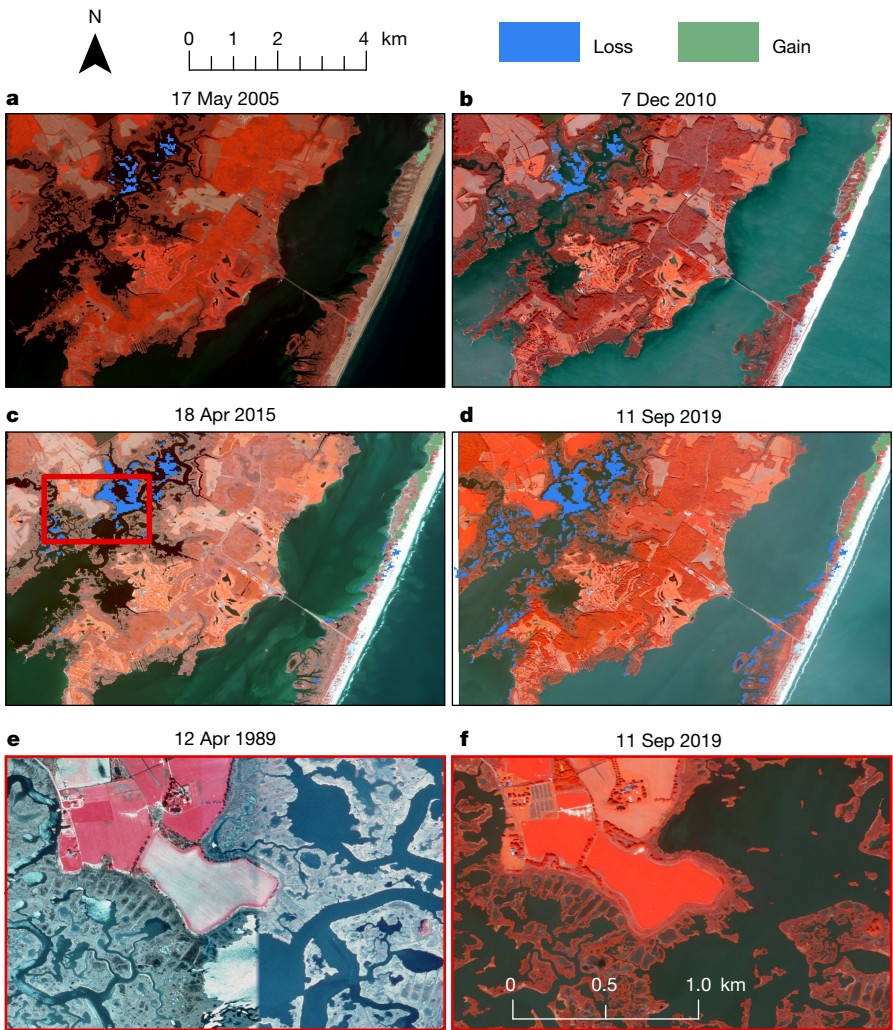

**Fig. 3 | Detailed salt marsh change for a section of the eastern shore of Maryland, USA. a**, Salt marsh change from 2000 to 2004. QuickBird image collected on 17 May 2005. **b**, Salt marsh change from 2000 to 2009. WorldView-2 image collected on 7 December 2010. **c**, Salt marsh change from 2000 to 2014. WorldView-2 image collected on 18 April 2015. Areas of increase were visible in this epoch along the north of the barrier island. **d**, Salt marsh change from 2000 to 2019. WorldView-2 image collected on 11 September 2019. **e**, An area of the salt marsh before losses. Orthoimage collected on 12 April 1989. **f**, Northeast area of complete loss and, to the southeast, an area of interior die-off, both identified as losses by the algorithm. WorldView-2 image collected on 11 September 2019. All images NIR, G, B in RGB. Created using ArcMap 10.8. © 2005, 2010, 2015, 2019 Maxar, NextView License.

important regional change driver in Oceania, in which the salt marsh–mangrove ecotone is altered as mangroves migrate poleward with increasingly warm conditions[8]. In this study, mangrove encroachment or misclassification corresponded with more than a third of gains in Australia. Mangrove encroachment increases carbon sequestration and the value of ecosystem services[33,34], but the full environmental and ecological impact is unclear. In the case of SLR adaptation, for example, low marshes were aggrading with 10 mm year$^{-1}$ of regional SLR[35], greater than a proposed 7 mm year$^{-1}$ threshold for mangroves[36]. In the case of restoration, salt marsh provides ecological structure quicker than mangroves[37]. Mapping the salt marsh–mangrove ecotone is challenging at the 30-m Landsat spatial resolution, and a combination of higher resolution and new methods are necessary to improve carbon and ecosystem monitoring in this ecotone. Our estimates of mangrove encroachment and misclassification were key for constraining estimates of salt marsh change in Oceania and illustrated the need to consider coastal systems as a whole to understand both blue carbon and changes to coastal resilience.

This study is a comprehensive salt marsh change analysis. Our change rates and associated uncertainties can improve carbon monitoring estimates in salt marsh ecosystems. Our revised salt marsh maps clarify

that these systems experienced a decline in net loss rates from 2000 to 2019. Salt marsh change was an order of magnitude lower than previous estimates, from −1.5 ± 0.5% year$^{-1}$ (ref. [38]) to −0.15 ± 0.01% year$^{-1}$. Recent global mapping of tidal wetlands found 90,800 km$^2$ of tidal marshes[39]. Using this extent and our change rates, we found mean carbon emissions from tidal marshes of 31.16 (0.067–63.95) Tg CO$_2$e year$^{-1}$, with the upper bound being similar to previous central estimates of tidal marsh emissions[38]. By comparison, using the change rates derived from the global tidal wetland map, a map of tidal flats, tidal marsh and mangroves[39], results in a much lower estimate of 5.19 (0.06–10.79) Tg CO$_2$e year$^{-1}$. We view the discrepancy between the two change rates as partly owing to the difference between monitoring tidal marsh condition versus cover. As seen in recent analysis of the Northeastern United States, far more change is happening to the marsh condition than to cover, but these condition changes do result in carbon loss[40]. Declining tidal marsh emissions suggests the success of efforts to protect and restore these ecosystems. The proximity of the lower bound of confidence to zero indicates the possibility that, with management and restoration, these systems could become net carbon sinks. However, the high uncertainty illustrates the need for further monitoring, mapping, carbon and soil-depth measurements.

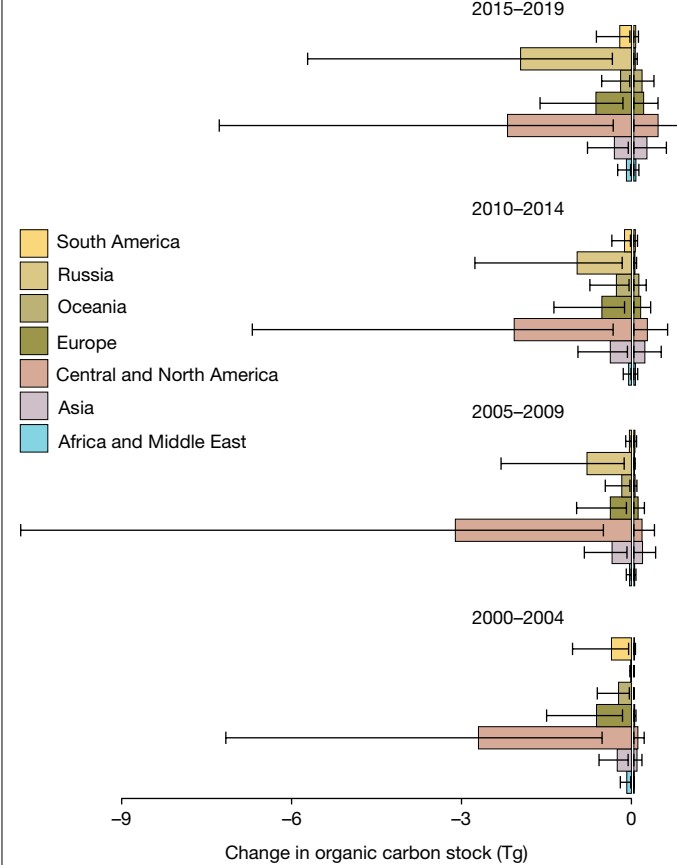

**2015–2019**

**2010–2014**

**2005–2009**

**2000–2004**

Legend:
- South America
- Russia
- Oceania
- Europe
- Central and North America
- Asia
- Africa and Middle East

Change in organic carbon stock (Tg)

**Fig. 4 | Change in SOCS.** Global change in SOCS within the first 30 cm attributable to salt marsh losses and potential gains as determined by carbon burial. Error bars represent 90% confidence intervals.

An incomplete global salt marsh extent adds uncertainty to our results. However, our use of the global tidal wetland change maps[39] upper extent demonstrates overlap between the confidence intervals of emissions estimates derived with their change rates and ours. Repeating our change analysis with the 2000 extent derived from the global tidal wetland map, a Landsat classification of mangroves, tidal marsh and tidal flats, is a priority. Future global salt marsh change analyses should continue to consider the surrounding area to provide insight into migration, persistence and interconnectedness of coastal ecosystems. Despite the uncertainty, our work addresses vital needs in blue carbon research, including improved mapping, uncertainty in carbon estimates and the effect of disturbance[41].

## Salt marsh carbon emissions

Globally, soil organic carbon stock (SOCS) losses were relatively consistent between epochs, despite the regional and temporal fluctuations of marsh loss. Conversely, gain was estimated with carbon burial rates, which increased linearly owing to the process being cumulative (Fig. 4). Total salt marsh losses from 2000 to 2019 represented a reduction of 4.88 (3.16–6.81) Tg C of aboveground biomass. Net loss represented a reduction of 0.045 (−0.14–0.115) Tg $CO_2$e year$^{-1}$ of carbon burial. The lower bound of net carbon burial represents the potential that current change already represents a net sink when considering only carbon burial. We evaluated net soil organic carbon loss to 30-cm depth with SoilGrids and in situ estimates, finding 19.0 (3.33–56.39) Tg C and 22.1 (−4.79–50.37) Tg C, respectively. New or frequently disturbed marshes probably have lower SOCS, therefore global values are expected overestimate SOCS loss in certain regions such as China and

Russia. Salt marsh loss can lead to increased $CO_2$ emissions from soils at depths >40 cm (ref. [42]). However, carbon accounting efforts offer varied assumptions about the depth of loss, with high-end scenarios varying from 90% to 100% SOCS loss[38], whereby using our in situ estimate a complete carbon loss in the first 1 m releases 73.99 (−15.24–167.4) Tg C. Owing to the high uncertainty of in situ soil organic carbon, zero is included in our confidence intervals; however, it is unlikely that loss results in SOCS gain. Because carbon accretion increases in regions with high regional SLR[43], the spatial relationships of coastal carbon make geospatial analysis critical for improved monitoring.

Salt marshes take time to accrete their extensive soil organic carbon stocks, with little difference in SOCS between recent marshes (1–15 years)[44]. Carbon accounting of anthropogenic loss processes can result in large reductions of SOCS in coastal wetlands, for example, a 57% reduction in SOCS of drained marshes[45]. In mangroves, the effect of disturbance type on sequestration and emissions has been explored[27]. A similar analysis of salt marsh emissions would further improve our understanding of blue carbon dynamics.

## Future monitoring and blue carbon needs

The United Nations Sustainable Development Goal 13 (Climate Action), a nonbinding call to integrate climate policy at the national level[46], and the UNFCCC Paris Agreement emission targets and progress reporting[47] are synergistic, providing exigence to addressing climate change and development in tandem[48]. Including blue carbon ecosystems in international policies is critical for achieving the Paris Agreement's Nationally Determined Contribution climate targets; however, their utility is complicated by uncertainty, as detailed in the Intergovernmental Panel on Climate Change (IPCC) AR6 report[49]. We demonstrate the feasibility of our approach for national to global monitoring of salt marshes by providing country-level estimates of salt marsh change, SOCS loss and 2019 SOCS (Supplementary Tables 1 and 12). Further country-specific in situ SOCS and biomass data could be incorporated to improve these estimates.

Although current marsh losses are lower than previous estimates, it is important to consider the extent of loss these ecosystems experienced historically. Coastal wetlands have lost approximately 46.4% of their area[10], with urban areas experiencing an even greater loss. For example, since 1776, Boston in the USA has lost 81% of its salt marsh[7]. Monitoring of this carbon-dense ecosystem is critical, as loss results in large $CO_2$ emissions per area. Restoration, migration, storms and SLR add uncertainty to how salt marshes will change in the future, complexity to blue carbon budgets and necessitate monitoring. Our monitoring approach allows for spatially explicit carbon accounting of salt marsh change that considers marsh loss, gain and recovery. Spatial estimates of change are necessary to understand the restoration potential of blue carbon systems and inform subnational monitoring.

Globally, 60% of salt marsh extent was at least partially protected[9] and Section 404 of the US Clean Water Act of 1972 protects wetlands connected to navigable water in the USA. However, these ecosystems were still changing, suggesting that reaching stability would require active management, including restoration and facilitated migration.

This work identifies several change hotspots in which more detailed analyses are necessary: (1) the Arctic, in which change was rapid, (2) the salt marsh–mangrove ecotone, (3) regions of expected anthropogenic-driven loss, (4) salt marsh regions affected by storm events and high rates of SLR, and (5) repeated globally for the new tidal marsh extent map[39]. This study provides sophisticated methods for monitoring these ecosystems and improves our understanding of carbon stocks and change. Our work is the most comprehensive effort to monitor salt marsh carbon globally, resulting in salt marsh change estimates, improved salt marsh carbon budget and an estimate of salt marsh recovery. We identified salt marsh change spatially across two decades, provided country-level estimates of carbon stocks informed

# Article

by local estimates of SOCS and identified hurricanes as a key driver of loss in the USA. Our results justify further monitoring to facilitate inclusion of the ecosystem in preservation-based carbon protocols and Nationally Determined Contributions to incentivize restoration and protection.

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

## Methods

### Change anomaly analysis

We conducted a global change analysis using the best available salt marsh extent maps[9]. When previous work compared salt marsh trends using a starting extent derived from Mcowen and ancillary data with higher-resolution regional mapping, a high correlation was found[32,36]. Although acknowledged as incomplete, those maps were the most comprehensive global salt marsh map available. We used the National Wetlands Inventory (NWI) extent in the USA owing to regional updates since the aggregation of the global extents. The salt marsh extent for each region was converted to a 30-m raster and uploaded to Google Earth Engine. We filtered the Landsat 5, 7 and 8 collections by 50% cloud cover, geometric accuracy, image quality and pixel quality to remove low-quality scenes and pixels. Our time series approach of comparing all quality images within an epoch mitigates the effect of any single image on the change outcome. All these process steps should mitigate the effect of the Landsat 7 shutter synchronization anomalies. We used Google Earth Engine to compare the salt marsh NDVI baseline from 1984 to 1999 and four 5-year periods from 2000 to 2019. These 5-year epochs of 2000–2004, 2005–2009, 2010–2014 and 2015–2019 allowed for a per-pixel analysis of change to minimize the effect of tidal stage. The baseline for Siberian watersheds was adjusted to 1984–2004 owing to the limited Landsat images in the region before 2000, resulting in three 5-year analysis periods (2005–2019).

NDVI has frequently been used to analyse salt marsh change[50,51] and estimate salt marsh biomass[52–54]. The link between remote sensing and ecological change is nonlinear and complicated but well established from the Arctic[55] to wetlands[17,56]. Tidal marsh monitoring has also been conducted with other indices[57,58] and several indices simultaneously[32]. Machine learning kernel approaches (kNDVI) has been proposed as a method to address saturation of NDVI[59]. However, many vegetation indices still use the relationship between red and near infrared. An intercomparison of the linear relationship of these indices, excluding kNDVI, and salt marsh aboveground biomass found that NDVI performed best[60]. An important concern in regional and global salt marsh change analysis is variability in leaf structure in these environments[58]. Regardless of the spectral index used, this would be an issue and something that future analyses can address with more in situ data and remote-sensing approaches.

To overcome some of these challenges, we applied a time-series NDVI anomaly approach to minimize the effect of seasonal and tidal variability and conducted a robust accuracy assessment protocol. We define anomalies as pixels that experienced greater than an absolute 0.2 magnitude of NDVI change, a threshold used in mangrove change analyses[17,18,61–63]. These methods and threshold have recently been used to understand storm impacts and loss in mangrove environments[63,64]. Our accuracy and threshold assessments further confirmed that this threshold had high user accuracy and successfully identified change with minimal inclusion of stable areas (Supplementary Tables 3–9). Salt marsh NDVI was compared during peak biomass months of August and September for the Northern Hemisphere and February and March for the Southern Hemisphere. The analysis unit was the watershed, HUC 6 watersheds within the USA and the World Wildlife Foundation (WWF) Basin 6 watersheds globally.

We calculated salt marsh anomaly metrics for four 5-year epochs from 2000 to 2019 using a combination of Python 3.8.10 and R 3.6.2. Salt marsh change metrics included recovery, salt marsh loss and gain anomalies, and loss, as well as gain anomalies within 100 m of the salt marsh. For a pixel to be considered recovered, it had to return to at least its reference period NDVI in a subsequent epoch. Salt marshes are unlikely to recover from certain loss processes such as herbivory[65], mangrove encroachment[8], eutrophication[11] and SLR[5]. However, recovery following losses from overwash[66], ice scour[67] and saltwater intrusion into freshwater environments[68] are all documented.

### Accuracy assessment

We conducted an accuracy assessment across 12,600 stratified random points, split into loss, gain and stable for each period (Supplementary Tables 3–6) and an accuracy assessment of salt marsh extent, including 6,845 stratified random points split between salt marsh and other land cover categories (Supplementary Table 7). We assessed recovery across 2,000 points split between loss and recovery (Supplementary Table 8). Each point was representative of the 30-m Landsat pixel in which it fell. We determined the status of a pixel with Google Earth Pro using historic imagery to determine change and ancillary imagery in the USA. Google Earth Pro imagery sources included Maxar, Airbus, the United States Geological Survey and NASA. The spatial resolution of these images varies but very-high-resolution (<3 m) imagery was used for most of our land cover verification. In limited instances, for example, Alaska, 30-m imagery was also used. Accuracy assessments included Alaska, Australia, Canada, China, Mexico, the UK and the USA to derive uncertainty estimates for each region. All locations had 1,800 points split between epochs, except Alaska with 1,344 and the USA with 2,256. We calculated confidence intervals using the accuracy assessment results[69]. The overall accuracy was 93%, 91%, 91% and 90% in 2000–2004, 2005–2009, 2010–2014 and 2015–2019, respectively (Supplementary Tables 3–6).

The anomaly analysis relies on a threshold to determine changed and unchanged pixels. All quality pixels are compared with the reference average, and the average across the epoch must exceed the 0.2 threshold to be considered changed, a threshold used in mangrove change analyses[17,18,62]. We further assessed the 0.2 NDVI threshold using an expanded accuracy assessment. We randomly selected salt marsh pixels with $a > 0.15$ magnitude change and determined whether we observed a change in the pixels using Google Earth Pro and available ancillary imagery. We found that, for loss thresholds, 0.2 and 0.19 had similar accuracy, with 0.2 having slightly higher accuracy. The 0.2 threshold was also slightly higher accuracy for gain areas and, therefore, we used a 0.2 threshold to best capture a pixel-wide change in vegetative extent (Supplementary Table 9). We further assessed our marsh loss within the USA relative to the Global Land Cover and Land Use 2019 dataset (GLAD 2019)[70]. Here we compared our loss pixels to the Land Cover Land Use (LCLU) class of that location in the GLAD 2019 data. We found that approximately 12% of our loss locations were sparse or non-emergent wetlands, and 73% were open water, showing a strong agreement between our change and the GLAD data.

### Mapping year

The mapping year, the year in which imagery was acquired, varied globally, and to verify the bias introduced by mapping year, we used beta regressions to assess the mapping year in the USA. The beta regressions compared percent change metrics (loss and gain) and mapping year (Supplementary Tables 10 and 11). North America has the largest concentration of mapped salt marsh of any continent. In the USA, the NWI is an irregularly revised mapping effort to track all USA wetlands. The NWI uses the Cowardin classification[71], consistent methods and varying aerial data sources[20]. These statistical analyses were limited to the USA owing to the availability of ancillary data and the metadata of the NWI. However, the analysis demonstrates that mapping date had a limited effect on the gains in 2000–2004 and the losses in 2015–2019. These change rates were used for our quantification of yearly emissions from salt marsh change.

The average mapping year varied regionally. For example, the USA had an average mapping year of 2007, compared with Alaska with an average mapping year of 1986. Mapping year also varied globally, with areas in Spain and Russia mapped in the 1980s. The mapping date most affected the 2000–2004 and 2015–2019 periods for loss and gain anomalies, respectively (Supplementary Tables 10 and 11).

Global salt marsh extent maps are lacking for many regions. However, a complete map of salt marsh would not change the result of this study markedly. For example, the salt marsh extent of India has recently been mapped, finding 290.49 km$^2$, approximately half a percent of global salt marsh[72]. Increased map accuracy and a uniform baseline mapping data are probably more beneficial to this analysis.

## Drivers of salt marsh change

Panel regression models were used to compare watersheds over time with LSLC, change within 100 m of the salt marsh, urbanization within the watershed and hurricane landfall and category or highest category in instances of several landfalls. The LSLC measure identifies periods of sea-level change that can be driven by factors such as ocean currents, climate cycles and storm events. The LSLC was derived from National Oceanic and Atmospheric Administration (NOAA) tide stations for each of the four 5-year periods. The 'rnoaa' package was used to download tide station data directly into R (ref. [73]). All stations were filtered to watersheds with salt marsh in the USA and by our study period, 2000–2019. All available data for each Center for Operational Oceanographic Products and Services (CO-OPS) station were split into the four epochs of our study, 2000–2004, 2005–2009, 2010–2014 and 2015–2019. Each period had a trend calculated by decomposing monthly mean sea level using the decompose function in base R, which uses a moving average to isolate trend, seasonality and error. The more complex seasonal-trend decomposition with Loess was not used owing to the recommendation that the season window is composed of at least seven time steps[74] and our use of monthly tidal data. Linear regression was then fit to the resulting trend estimating change per year in millimetres. Watersheds with several tide stations were averaged, resulting in a single local sea-level trend for each watershed. The study included 72 watersheds within the USA, of which 45 had tide station records and which we used in the panel analysis.

Hurricane track and intensity data (HURDAT2) were acquired from National Weather Service and processed using the R package 'tidyverse'[75,76]. We processed HURDAT2 data for both the Atlantic and Pacific oceans, but no Pacific typhoons affected watersheds within the USA. We imported the processed HURDAT2 data as a delimited text layer into QGIS 3.12.263, creating a buffer surrounding each point based on the hurricane diameter[77]. If the hurricane diameter was missing, we used the average hurricane diameter for the corresponding category of storm. We used these data to determine which watersheds were affected in each epoch and the highest category of hurricane impact.

We determined watershed urbanization using a global map of impervious surface increases from 1984 to 2018 derived from the Landsat archive[78]. We calculated the amount of urbanization for each watershed using the zonal histogram tool in QGIS 3.12.2. Annual impervious surface estimates were aggregated to the study periods quantifying total artificial impervious surface added in each period.

## Carbon monitoring

We estimated the impact of salt marsh change on ecosystem carbon, including aboveground carbon, carbon burial and soil organic carbon. Salt marsh loss was considered the complete loss of aboveground biomass, carbon burial and SOCS. These losses were calculated for both 30 cm and 100 cm to cover a range of loss estimates. Tidal wetland carbon monitoring has previously been conducted with regional values and ecosystem extents[79]. Aboveground biomass was estimated from plots across the USA from Byrd et al. (705.9 ± 720 g m$^{-2}$ (standard deviation)) with a carbon conversion of 0.441 (refs. [58,80]). Belowground biomass was assumed to be included within soil core measurements of SOCS and was not computed separately. We estimated carbon burial using the latest values from the literature of 168 ± 7 g C m$^{-2}$ year$^{-1}$ (ref. [81]), which is lower than older estimates of carbon burial (218 ± 24 g C m$^{-2}$ year$^{-1}$) (ref. [1]). Carbon burial rates were used to calculate carbon increases from gains in salt marsh extent. SOCS was extracted from the SoilGrids250m

dataset[82]. These estimates were averaged by change type (loss or gain), epoch and watershed. Consistent with previous work, we assume a complete loss of SOCS, which has been estimated to take place over years but still underestimates total carbon lost[38]. These spatial estimates of SOCS loss were compared with globally derived estimates of SOCS from the Coastal Carbon Research Coordination Network (CCRCN) and the literature[83,84]. Our in situ estimate derived from the CCRCN was 270.4 ± 2.8 Mg ha$^{-1}$ (ref. [83]), which was slightly lower than values from the literature (317.2 ± 19.1 Mg ha$^{-1}$)[84]. The Coastal Carbon Data Clearinghouse values were exclusively within North America and Europe (Supplementary Table 13). The use of 1 m to calculate SOCS ignores another source of notable uncertainty, which is soil depth, for example, the mean depth of cores representing deposit depth in emergent vegetation was 194.5 cm (Supplementary Table 13). In comparison, the SoilGrids dataset accounts for spatial variation but underestimates the carbon lost.

The upper bound of tidal marsh extent was estimated using the recent mapped extent of 90,800 km$^2$ (ref. [39]). Salt marsh carbon estimates and our change rates were applied to this extent of tidal marsh. Previous blue carbon budgets have used an upper bound of tidal marsh, which included mangroves and marshes[1,85,86].

Previous budgets estimated partial losses of SOCS[1]. We offer a total loss that assumes an eventual complete loss of carbon. Carbon gain is calculated by combining carbon burial and aboveground carbon. The $CO_2$ emission estimates of soil carbon change used yearly change rate for loss and gain from the epochs 2015–2019 and 2000–2004, respectively. These epochs were the least affected by mapping year. We propagated error throughout the analysis with 100,000 Monte Carlo simulations for all confidence intervals. Uncertainty reported in parentheses is 90% confidence intervals or standard error after a ±.

## Data availability

The data that support the findings of this study are openly available at https://doi.org/10.3334/ORNLDAAC/2122. Loss and gain maps are available at https://mangrovescience.earthengine.app/view/salt-marshchange and https://mangrovescience.earthengine.app/view/saltmarshsoc.

## Code availability

Examples of the code used to process the data are available at https://github.com/campban/Global_saltmarsh.

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

**Acknowledgements** This research was supported in part by the NASA Carbon Monitoring System programme (grant number 16-CMS16-0073). A.D.C. was supported by the NASA Postdoctoral Program Fellowship administered by Oak Ridge Associated Universities. We would also like to thank A. Stovall and C. Doughty for reading and offering edits on an early version of the paper. Maxar data were provided under the National Geospatial-Intelligence Agency's NextView license agreement.

**Author contributions** A.D.C. and L.F. conceived and designed the experiments. D.L., L.F. and L.G. contributed analysis tools. A.D.C. adapted and expanded the analysis tools for salt marsh environments. A.D.C., L.F. and D.L. analysed the data. A.D.C. wrote the first draft of the manuscript, with input from L.F. A.D.C., L.F., D.L. and L.G. revised the manuscript.

**Competing interests** The authors declare no competing interests.

**Additional information**
**Correspondence and requests for materials** should be addressed to Anthony D. Campbell.

**Extended Data Table 1 | Salt marsh gains and losses in km² for each 5-year epoch by region**

| Type | Region | 2000-2004 | 2005-2009 | 2010-2014 | 2015-2019 |
|---|---|---|---|---|---|
| Loss | Africa & Middle East | 13.7±1.8 | 6.4±0.8 | 9.1±1.2 | 14.6±1.9 |
| | Asia | 57.8±7.6 | 76.5±9.7 | 78.8±10.4 | 65.5±8.4 |
| | Central & North America | 312.1±41.1 | 368.9±47.0 | 259.2±34.3 | 273.3±36.4 |
| | Europe | 97.0±12.8 | 56.2±7.2 | 86.3±11.4 | 103.2±13.3 |
| | Oceania | 37.1±4.9 | 33.0±4.2 | 53.0±7.0 | 37.7±4.8 |
| | Russia | 1.2±0.2 | 112.1±14.3 | 133.8±17.7 | 279.4±35.9 |
| | South America | 81.6±10.7 | 8.2±1.0 | 24.2±3.2 | 43.7±5.6 |
| Gain | Africa & Middle East | 3.4±0.9 | 14.5±3.0 | 15.1±2.8 | 10.3±1.8 |
| | Asia | 66.9±17.9 | 115.9±23.9 | 48.8±9.0 | 45.6±8.0 |
| | Central & North America | 85.1±22.7 | 88.6±18.3 | 113.2±21.0 | 222.7±39.1 |
| | Europe | 17.7±4.7 | 71.5±14.8 | 53.7±9.9 | 64.2±11.3 |
| | Oceania | 0±0 | 26.1±5.4 | 79.7±14.8 | 68.5±12.0 |
| | Russia | 0±0 | 11.4±2.3 | 12.0±2.2 | 5.6±1.0 |
| | South America | 13.2±3.5 | 9.8±2.0 | 8.2±1.5 | 8.1±1.4 |