## [Peer Review File · Nature]

Manuscript Title: Global hotspots of salt marsh change and carbon emissions.

Reviewer Comments & Author Rebuttals

Reviewer Reports on the Initial Version:

Referees' comments:

Referee #1 (Remarks to the Author):

Global hotspots of salt marsh change and carbon emissions 2021-10-16288

This study produced a global change analysis of salt marsh loss, gain, recovery and carbon stock changes with Landsat satellite data and Google Earth Engine. Given that salt marshes sequester more carbon per unit area than most other ecosystems, this is an important undertaking, especially since salt marsh change maps are not available for many countries. The authors use a simple NDVI vegetation index thresholding technique to classify satellite imagery into marsh gain, loss and no-change categories. It is not clear how the thresholding defined gains vs. losses. They use as an initial base map a UN global map of salt marshes which represents the best available global dataset, though it is based on an assemblage of different datasets from different dates. The authors do a good job of testing the sensitivity of the NDVI threshold and its resulting map accuracy, as well as the influence of the date of the initial salt marsh base map used in the time series analysis to detect change. However the authors show in analyses in the supplement the inconsistencies of using an NDVI threshold to detect change, where in some cases an increase in NDVI results in marsh loss and conversion to another land use like agriculture. They made corrections at the regional level to correct these inconsistencies. However I'm surprised they did not use a more complex mapping algorithm available in Google Earth Engine like machine learning for example that can take multiple input variables and therefore produce accurate maps in a range of different environmental settings. The accuracy assessment seems robust, with thousands of points used to test accuracy. Accuracy was high, but the methods were not described well enough to have confidence in the approach. The authors said they used high spatial resolution imagery in Google Earth Pro, which is a standard technique – but did not say what the spatial resolution of the imagery was, or what kind of imagery was used. The imagery would need to be pretty fine scale, $\leq 3\text{m}$ to visually detect marsh loss or gain. Also areas with large amounts of detected change, like Russia, may or may not have that high resolution imagery available within Google Earth Pro. Were there many accuracy assessment points within these area of high change? In general the manuscript writing can be improved. The first section reads like a series of statistics, and the following section is not well organized. The paper is also interspersed with some unnecessary details about specific areas and policies that could be removed to streamline the writing, especially given that this is a global mapping effort. Reporting of error terms is not always consistent or well defined. Detailed comments are below.

Lines 57 and 59: Error terms are reported, but what are they? Confidence intervals?

Figure 1. This is overall a very good figure – the line plots by latitude and longitude are effective. Font size in figure 1a is much too small. Bar charts in figures b-h should have error bars. There are error bars in figure j but they are not defined.

Line 73: A net loss is reported without error terms, though this is based on gain and loss estimates that do have error terms – these should be propagated to give a range around the net loss value.

Line 90: Sentence starting, "Ice scour.." belongs better in the marsh drivers of change section. In general the paper should be edited so sentences about drivers of change are organized together in

its own section.

Line 108: Error terms here and throughout not defined. Are these confidence intervals?

Line 116 – 119: This section on SOCS is hard to follow – not enough detail is given to understand the where the area being discussed is located or why it was a resilient region.

Line 129 – 138: This section is very hard to follow and to verify against the numbers in Figure 3. It is hard to tell if the actual or revised extent and emission values are what is being reported as the final values.

Line 144: This section on Marsh recovery, change drivers and landward migration needs to be fully edited for better organization. These are three separate topics but they are not clearly separated here.

Line 158: Sentence starting “Two years after..” seems extraneous and too detailed for a paper on a global mapping effort and should be removed.

Figure 4: The error bars in the bar charts have not been defined.

Line 170: Define the LSLC acronym.

Line 176: What is meant by “more impactful” here?

Line 182: run-on sentence

Line 209: Add “USA” to Boston

Line 217: Statement “The Clean Water Act of 1972 protects all wetlands” is not true. It only protects wetlands adjacent to navigable waterways.

Lines 219 – 223 are hard to follow, are not logical, and should be dropped.

Line 254: Define the NDC acronym.

Methods

Line 262: How did you address the line striping in Landsat 7?

Line 265: Why is NDVI the most appropriate vegetation index? Many more have been found to be much more effective for mapping wetland vegetation, like the wide dynamic range vegetation index for example, or soil adjusted vegetation index.

Line 270: What is meant by minimal inclusions?

Line 277: It is unclear how a gain and a loss anomaly was defined. Is an increase in NDVI considered a gain?

Line 288: The imagery within Google Earth Pro and other ancillary imagery types should be reported, including the spatial resolution. Authors should also report if assessment points covered areas of substantial change according to the time series maps.

Line 303: Use of beta regressions here seem appropriate given the dependent variable is percent change

Line 312: The base map for Russia was in the 1980s. It is unclear how this early date influenced

mapping results, though little change was reported in the 2000-2004 epoch but substantial change was reported in later epochs. Potentially there was marsh change from 1980 to 2000 that does not get reported.

Line 320: Panel regressions are an appropriate method here given this is time series data for a series of watersheds.

Line 337: What is Hurdat2 data?

Line 351: Salt marsh carbon stocks are not considered ecosystem services, though carbon sequestration is a service

Supplement:

General comment: define all acronyms in table and figure captions.

Table S3: The caption should be rewritten to more accurately explain the table. "Reference" and "Salt Marsh Change" are not classes.

Table S7: What year is this salt marsh extent accuracy assessment for?

Sections on China reclassification with ancillary data and Oceania mangrove encroachment analysis: These sections show the inconsistencies of just relying on a simple single vegetation index threshold for a global classification. Other indices could have been included in the algorithm to make it more robust to varying conditions like these described here.

Last sentences in section on Change in the marsh and neighboring area – how can erosion increase salt marsh resilience?

Table S15: It would make sense to report values in Teragrams here. Are these annual values?

Referee #2 (Remarks to the Author):

Review of Global Hotspots of Salt Marsh Change and Carbon Emissions

I enjoyed reading Global Hotspots of Salt Marsh Change and Carbon Emissions. In my opinion this work is an important contribution to blue carbon science because most change analyses to date have focused on mangrove ecosystems which have distinct advantages from a remote sensing perspective. Although my expertise is not in remote sensing or change analysis the methods applied appear to be robust.

Major Comments

1. The change analysis provides several important and high-impact insights. Hotspots of losses in the US Gulf Coast are a well-documented phenomenon lending confidence in the methods used here. Less well understood are the losses in Alaska and Russia, which lead the authors to identify the Arctic as a large knowledge gap (L245). The epoch-based analysis provided insights on the causes of marsh loss and recovery, such as the relationship between loss and hurricane strength followed by recovery in North America. Change analysis is the primary strength of the manuscript and was appropriately placed up front.

2. The consequences of area change for carbon gains and losses are also a major emphasis of the work, which in my opinion is less rigorous and detracts from a complete discussion of the area change analysis. For example, a more thorough discussion of saltmarsh loss and gain in China seems important considering that that China lost globally significant areas of saltmarsh during the

time period this analysis covers. China is discussed a bit in the supplementary material with respect to classification, but there is little discussion related to the so-called "Great Seawall of China" that constitutes (I assume) most of the direct human impact on coastal marshes since the later part of the 20th century; direct land use conversion is also the dominant cause of mangrove losses. By comparison, saltmarsh losses in the US, Europe, and elsewhere have been driven by less-direct human impacts such as sediment starvation, accelerated sea level rise, and coastal erosion. I suggest the authors invest more text in this short-format article to global spatial patterns in the causes of loss and gains of salt marsh area, and less text to carbon consequences. I am sure their extensive knowledge on this subject would be enlightening.

I trust the authors on their analysis but have seen the rate marsh reclamation reported as 40,000 ha/yr from 2006-2010 (Ma et al. 2016, Science); this was an opinion piece and lacked a citation but the rate is far higher than reported here (7,602 ha over 5 years) for China in Table S1. Given some of the issues with misclassification in China discussed in the supplementary material, some discussion about how the current change analysis compares to previous regional estimates seems in order.

As a minor note, I know the term "reclamation" is used to describe conversion of coastal wetlands to agriculture, but the term carries a negative value judgement about coastal wetland ecosystem services that is counter to that in this paper. I recommend using a different term such as "drainage".

Another topic that is discussed only briefly in the paper and the supplement, but could use more focus, is the encroachment of mangroves into saltmarshes.

3. The implications of area gains and losses for carbon stocks is a reasonable goal of such a paper and an expectation of readers as a logical next step. However, in my opinion the emphasis on carbon estimates is larger than warranted by the reduction in uncertainty over previous estimates. I appreciate that the authors acknowledged uncertainty in the global/regional area of coastal marshes when calculating stock change. I agree. Given the large losses in Russia and the poor representation of Northern Russia in the global map of salt marshes used in this study (along with Canada, South America, and Africa) the lack of good maps is reason enough for caution.

The authors used a spatial database – SoilGrids -- for soil carbon estimates. The metadata for SoilGrids states that "Tropics, wetlands, semi-arid to hyper-arid areas and mountains are still largely under-represented.". It was not clear from the supplementary information how good the database is for tidal wetlands, but it is unlikely that SoilGrids represents the best-possible source of such data. I am curious why the authors did not choose to use the Coastal Carbon Atlas which has over 6,000 soil profiles from tidal wetlands globally. There may be good reasons such as the fact that the Atlas is biased towards the US, but it also has a lot of data from outside the US. In any, case the current paper does not seem to be using the best-available soil carbon stock data. One advantage of the Atlas is that it would allow a more precise calculation of soil carbon stocks to 1 meter as the data are disaggregated by depth. Similarly, rates of soil carbon accretion were taken from an older source (published in 2013) while more recent and extensive compilations are available in the Coastal Carbon Atlas database.

I suggest that the authors keep the change in carbon gain in Fig 1, delete Fig 3 (which is premature), move the carbon discussion to near the end of the paper to reduce emphasis, and dedicate space to discussions that are more squarely based on area gains and losses.

Minor Comments

1. It appears from the supplementary information that the authors estimated belowground biomass and added it to soil carbon stocks. This is problematic because soil carbon stock estimates in these systems typically include roots already (i.e. roots are rarely separated from soil cores when quantifying stocks). Thus, there may be double counting. The error is probably relatively

small compared to the soil organic carbon itself, but it should be corrected. An example of where this issue was taken into account is in Rogers et al. (2019, Nature).

2. Many of the comparisons used to help readers understand the scale of changes were not compelling . I have some idea of the area of Hong Kong but no idea about the area of it's territorial waters which could be far larger (L71). I realize that salt marshes are essentially marine grasslands but I am unclear what to conclude from comparing carbon losses in these systems to terrestrial grasslands (L107). The comparison to Indonesian oilseed production for Chinese consumption seems a little odd.

L37. The gap was never stated. Add a sentence to L36 that clearly states the gap to be filled.

L57. I suggest "...), which is similar to the lower bound and an order of magnitude..." to provide a more balanced assessment.

L90. The authors should be cautious about some of their interpretations. I agree that there is ice in the image of the Maryland eastern shore but am quite sure that ice scour is not an important process there as it is in the northeastern US. That area does not freeze long enough to cause scour.

L112. Another good citation for the depth of disturbance is Kauffman, J.B., Heider, C., Norfolk, J. and Payton, F. (2014), Carbon stocks of intact mangroves and carbon emissions arising from their conversion in the Dominican Republic. *Ecological Applications*, 24: 518-527. <https://doi.org/10.1890/13-0640.1>

L129. The sentence "including salt marshes and other tidal marshes" raises questions about what types of tidal marshes were in or out of the current analysis. What are these other tidal marshes and were they included in the change analysis?

L157. What evidence is there that restoration and natural revegetation are the causes of the apparent recovery rate?

L186. Please find a more precise term in place of "vibrancy"

L188. It is not clear how this analysis detected that a site "got wetter". Please use more precise language to help the reader understand the inference.

General Comments

This paper presents initial results from a global scale analysis of changes as gain and loss, in saltmarsh(?) inferred from changes in a vegetation index derived from the Landsat series of satellites (2000-2019), with change identified from within a historic saltmarsh boundary from Mcowen et al. (2017) in four epochs 2000-2004, 2005-2009, 2010-2014, 2015-2019. The changes were used to estimate changes in Carbon stocks at local to regional scales, providing figures that were close to, but lower than current best case estimates. The results were validated from a global series of observation points. More detailed assessment of the potential drivers of change were conducted for the continental United States using land-cover data sets.

As a whole the paper does represent a potentially useful starting point to the growing body of work that is using global satellite archives and on-line data stores and processing environments to map and measure changes in the extent and properties of a range of environments, e.g. forests, inter-tidal areas, coastlines, mangroves, seagrass, coral reefs (Runting et al., Crowther et al., Murray et al., Galiatsatos et al., Mao et al., Lyons et al.,). There are several major concerns with the content and focus of the paper, outlined below, that need to be addressed before it is in a form suitable for publication. Once these are addressed this will be a highly significant contribution to our ability to map and monitor changes in this essential ecosystem and to provide more accurate spatial data to drive C-stock and budget assessments from local to national and global scales.

Supporting references:

Mcowen C, Weatherdon L, Bochove J, Sullivan E, Blyth S, Zockler C, Stanwell-Smith D, Kingston N, Martin C, Spalding M, Fletcher S (2017) A global map of saltmarshes. *Biodiversity Data Journal* 5: e11764. <https://doi.org/10.3897/BDJ.5.e11764>

Runting, R.K., Phinn, S., Xie, Z., Venter, O. and Watson, J.E., 2020. Opportunities for big data in conservation and sustainability. *Nature communications*, 11(1), pp.1-4.
Hanson et al.

Crowther, T.W., Glick, H.B., Covey, K.R., Bettigole, C., Maynard, D.S., Thomas, S.M., Smith, J.R., Hintler, G., Duguid, M.C., Amatulli, G. and Tuanmu, M.N., 2015. Mapping tree density at a global scale. *Nature*, 525(7568), pp.201-205.

Galiatsatos, N., Donoghue, D.N., Watt, P., Bholanath, P., Pickering, J., Hansen, M.C. and Mahmood, A.R., 2020. An assessment of global forest change datasets for national forest monitoring and reporting. *Remote Sensing*, 12(11), p.1790.

Murray, N.J., Phinn, S.R., DeWitt, M., Ferrari, R., Johnston, R., Lyons, M.B., Clinton, N., Thau, D. and Fuller, R.A., 2019. The global distribution and trajectory of tidal flats. *Nature*, 565(7738), pp.222-225.

Mao, Y., Harris, D.L., Xie, Z. and Phinn, S., 2021. Efficient measurement of large-scale decadal shoreline change with increased accuracy in tide-dominated coastal environments with Google Earth Engine. *ISPRS Journal of Photogrammetry and Remote Sensing*, 181, pp.385-399.

B. Lyons, M., M. Roelfsema, C., V. Kennedy, E., M. Kovacs, E., Borrego-Acevedo, R., Markey, K., Roe, M., M. Yuwono, D., L. Harris, D., R. Phinn, S. and Asner, G.P., 2020. Mapping the world's coral reefs using a global multiscale earth observation framework. *Remote Sensing in Ecology and Conservation*, 6(4), pp.557-568.

Specific questions to address:

A. *Summary of the key results*

This paper presents initial results from a global scale analysis of changes as gain and loss, in saltmarsh(?) inferred from changes in a vegetation index derived from the Landsat series of satellites (2000-2019), with change identified from within a historic saltmarsh boundary from Mcowen et al. (2017) in four epochs 2000-2004, 2005-2009, 2010-2014, 2015-2019. The changes were used to estimate changes in Carbon stocks at local to regional scales, providing figures that were close to, but lower than current best case estimates. The results were validated from a global series of observation points. More detailed assessment of the potential drivers of change were conducted for the continental United States using detailed land cover data sets.

B. *Originality and significance: if not novel, please include reference*

This paper is original in terms of the problem addressed, data sets used and methods applied to estimate changes in salt-marsh. The paper represents a highly useful starting point for developing a more accurate approach to mapping and measuring changes in saltmarsh extents, from local to global scales. The derivation of the research problem cites appropriate literature, as does the development and implementation of the methods used, along with the analysis and interpretation of the results. There are some limitations with the literature covered in relation to similar locally detailed, global scale change analyses, and the satellite vegetation indices used. These limitations are outlined in the detailed comments table below.

C. *Data & methodology: validity of approach, quality of data, quality of presentation*

This paper

Data – appropriate data and methods were used through the paper, and their selection and limitations are clearly stated and linked to appropriate literature. More details are needed on some of the data sets used (e.g. validation of land cover changes) and selection of satellite index data sets, with details outlined in the table below.

Methodology – appropriate and logical design, supported by relevant literature in their application and assessment of limitations. Several sections, as noted in the comments table below, do require expansion and clarification of the methods used and their limitations, especially validation reference data, and in some cases should be considered for removal to simplify the message of the paper (e.g. detailed analysis within US).

D. *Appropriate use of statistics and treatment of uncertainties*

This paper does appear to use statistics and uncertainties in the mapping, saltmarsh area change and C-stock calculations.

E. *Conclusions: robustness, validity, reliability*

The conclusions reached in this paper are primarily supported by the data sets, methods and interpretation methods. However, the scope of the findings needs to be clarified and then modified to fit within the data sets and analysis methods used, noting the reliance on a sub-optimal (but best on offer) global salt marsh extent compilation, and that this is the first step in developing an approach able to be used for saltmarsh change

analysis. Once this is done it, then the work can be shifted to assessing saltmarsh C-dynamics.

F. *Suggested improvements: experiments, data for possible revision*

See list of requested changes in the “Specific Comments” table below.

G. *References: appropriate credit to previous work?*

Yes – with some required changes, see “General Comments” above and requested changes in the “Specific Comments” table below.

H. *Clarity and context: lucidity of abstract/summary, appropriateness of abstract, introduction and conclusions*

The paper is constructed and written clearly, and is free from grammatical and punctuation errors. The abstract is appropriate but requires several edits, see requested changes in the “Specific Comments” table below. The introduction and conclusion also require specific changes as noted in the “Specific Comments” table below.

\

Specific Comments to be addressed:

Line number	Comment
15	This is very abstract for the majority of readers - and Singapore has island and mainland sectionsprovide actual area in km^2 ...maybe from https://www.worldatlas.com/features/countries-by-area.html#countriesBySize ?
19-20	Suggest results stay within context of the data and methods and also recognise that multiple countries and private companies are already doing this. It would be better to place this work in context of those approaches and where they are going.
21-22	The findings would also be strengthened if placed in context of other estimates of global; saltmarsh extent and C dynamics e.g. Alongi, D. (2020) Carbon Balance in Salt Marsh and Mangrove Ecosystems: A Global Synthesis. J. Mar. Sci. Eng. 2020, 8(10), 767; https://doi.org/10.3390/jmse8100767
35-36	This would be significantly improved by starting with "saltmarsh extent" as that is a primary data product that can then be used and assessed, and collectively improved. e.g. see the global inter-tidal and global coral reef data sets.

39-40	Can you clarify in more detail what this means, especially the 'within 1.8km of the previously....' in "We processed all Landsat 5, 7, and 8 imagery with Google 39 Earth Engine within 1.8 km of the previously mapped global salt marsh extent9 "
41-42	So the Mcowen et al. (2017) polygons were used as the "baseline" extent and the assumption was that if the amount of vegetation changed , the NDVI value would change, i.e. anomaly detection? Why not just state that method more clearly? It also seems subject to significant bias:  - extent of Mcowen et al. polygons; - assumption that saltmarsh gain or loss is shown by change in NDVI Doesn't this only show the change in NDVI valuesbut no real link to actual saltmarsh dynamics ? Case for this still needs to be made more clearly.
47	No need to overcomplicate things.... why not just say "mainland continental US, excluding Alaska, HI and others..."
59-60	Keep to single area unit - not ha - use km². This sentence needs more detail - as it is not clear what it means ?
62	Insert - at appropriate local level of details to detect salt-marshes and their changes.
78	coastal erosion rates ? Can you provide references to back up this assertion e.g. in terms of global shoreline change e.g. Mao, Y., Harris, D.L., Xie, Z. and Phinn, S., 2021. Efficient measurement of large-scale decadal shoreline change with increased accuracy in tide-dominated coastal environments with Google Earth Engine. ISPRS Journal of Photogrammetry and Remote Sensing, 181, pp.385-399.
Figure 2 – 94-101	This will be easier to interpret if the size of all image maps was maximized, and if each image panel was labelled with the time period it represents.
120	Isn't this a fundamental limitation of the approach - and constrains the scope of findings to looking at change from that data set as a baseline?
144-195	It is still very unclear how the methods and data used are able to accurately map saltmarsh recovery and identify landward migration (isn't this limited by the Mcowen polygons) andvery difficult to conceptualize how change drivers can be inferred, let

	alone assigned with any confidence ?
182-195	This is extremely interesting - can we do it globally with https://esa-worldcover.org/en or https://livingatlas.arcgis.com/landcover/
237-242	Just a comment – this is a very realistic and useful assessment
265-266	Re.NDVI Not really - the paper by Lopes et al. 2020 only covers a small saltmarsh area in Portugal - given the many challenges and limitations of NDVI with bright and dark soils, open canopies and soil moisture, there really should be more careful consideration of a more effective index?
284-287	How were the "points" reference data that could be used to validate the saltmarsh presence or absence - what is their physical link to the variable mapped, and how do they match in space and time?
301 -318	This is very confusing - and i am not sure how you would be able to explain this a group of decision makers to give them confidence in the mapping and ability to identify gains and losses accurately ? Can this be improved significantly?
319-349	On reflection this section of analyses and interpretation may be better served in a separate paper as it distracts and confuses the (already complicated) global story of saltmarsh gain/loss and C budgets.

END OF REVIEW

Author Rebuttals to Initial Comments:

Referees' comments:

Referee #1 (Remarks to the Author):

Global hotspots of salt marsh change and carbon emissions 2021-10-16288

This study produced a global change analysis of salt marsh loss, gain, recovery and carbon stock changes with Landsat satellite data and Google Earth Engine. Given that salt marshes sequester more carbon per unit area than most other ecosystems, this is an important undertaking, especially since salt marsh change maps are not available for many countries. The authors use a simple NDVI vegetation index thresholding technique to classify satellite imagery into marsh gain, loss and no-change categories. It is not clear how the thresholding defined gains vs. losses. They use as an initial base map a UN global map of salt marshes which represents the best available global dataset, though it is based on an assemblages of different datasets from different dates. The authors do a good job of testing the sensitivity of the NDVI threshold and its resulting map accuracy, as well as the influence of the date of the initial salt marsh base map used in the time series analysis to detect change.

However the authors show in analyses in the supplement the inconsistencies of using an NDVI threshold to detect change, where in some cases an increase in NDVI results in marsh loss and conversion to another land use like agriculture. They made corrections at the regional level to correct these inconsistencies. However I'm surprised they did not use a more complex mapping algorithm available in Google Earth Engine like machine learning for example that can take multiple input variables and therefore produce accurate maps in a range of different environmental settings. The accuracy assessment seems robust, with thousands of points used to test accuracy. Accuracy was high, but the methods were not described well enough to have confidence in the approach. The authors said they used high spatial resolution imagery in Google Earth Pro, which is a standard technique – but did not say what the spatial resolution of the imagery was, or what kind of imagery was used. The imagery would need to be pretty fine scale, $\leq 3\text{m}$ to visually detect marsh loss or gain. Also areas with large amounts of detected change, like Russia, may or may not have that high resolution imagery available within Google Earth Pro. Were there many accuracy assessment points within these area of high change? In general the manuscript writing can be improved. The first section reads like a series of statistics, and the following section is not well organized. The paper is also interspersed with some unnecessary details about specific areas and policies that could be removed to streamline the writing, especially given that this is a global mapping effort. Reporting of error terms is not always consistent or well defined. Detailed comments are below.

Response by Author

We thank the reviewer for their detailed and beneficial review of the manuscript. We have addressed all reviewer concerns below in detail.

We especially highlight that our approach has been verified across many studies. We agree that a machine learning approach to mapping with multiyear composite classifications of salt marsh would provide additional information and insight into the change and stability of these ecosystems. Our method however is addressing a different question – that of change within the system. We do produce a replacement for such a map. Furthermore, regionally, these maps exist (for example in the USA and China¹), but a global analysis is complicated due to tidal effects, variation in salt marsh vegetation type and species, and training data.

In response to the reviewer's comments, we have added an additional literature review^{2,3,4,5,6,7,8,9,10,11,12,13,14,15,16,17,18} and included machine learning kernel¹². However, other vegetation indices SAVI, WRNDVI, and kNDVI are primarily focused on addressing saturation, a frequent issue with NDVI in forested environments but less frequent in tidal marshes. Our preliminary evaluation of a machine learning kernel made change detection more difficult in salt marsh environments.

In response to the reviewer's comments, we have gone through the manuscript and clarified all error terms.

We have also addressed the reviewer's concerns by providing more detail about the regions from which accuracy points were derived and added additional detail on our accuracy assessment approach.

The introduction was rewritten to deemphasize statistics and increase clarity and narrative flow. The manuscript was reorganized, moving the carbon monitoring section to the end of the paper and making several other changes suggested by reviewers.

Detailed reviewer responses:

Lines 57 and 59: Error terms are reported, but what are they? Confidence intervals?

We updated error terms, all 90% confidence intervals are in parentheses and all SE is reported with a plus or \pm . This clarification was added to the methods.

Figure 1. This is overall a very good figure – the line plots by latitude and longitude are effective. Font size in figure 1a is much too small. Bar charts in figures b-h should have error bars. There are error bars in figure j but they are not defined.

Figure 1 has been split into a map and change figure. Both have been changed to km². We have enlarged text in the updated figure 1-2 allowing for additional clarity. Error bars were added and defined in the caption.

Line 73: A net loss is reported without error terms, though this is based on gain and loss estimates that do have error terms – these should be propagated to give a range around the net loss value.

We addressed this by propagating the loss/gain error with monte carlo simulations. This results in a CI 90% of 1452.84 (733.1-2172.07 km², 90% Confidence interval).

Line 90: Sentence starting, “Ice scour..” belongs better in the marsh drivers of change section. In general the paper should be edited so sentences about drivers of change are organized together in its own section.

Thank you – We have entirely removed the reference to ice scour and refocused our results on describing the loss from SLR in response to Reviewer 2’s comments. As suggested, we have reorganized the text, moving discussion and descriptions of change to the change drivers section (Lines 128-154).

Line 108: Error terms here and throughout not defined. Are these confidence intervals?

We have added text stating these are confidence intervals.

Line 116 – 119: This section on SOCS is hard to follow – not enough detail is given to understand the where the area being discussed is located or why it was a resilient region.

We have removed parts of this section instead comparing SoilGrid to *in situ* measurements of soil organic carbon stock¹⁹. And discussing both these data sources relative to values from the literature for SOCS²⁰ and carbon burial²¹.

Line 129 – 138: This section is very hard to follow and to verify against the numbers in Figure 3. It is hard to tell if the actual or revised extent and emission values are what is being reported as the final values.

We have revised the section for clarity focusing on our contribution to blue carbon accounting. Figure 3 as been removed based on the recommendation of reviewer 2.

Line 144: This section on Marsh recovery, change drivers and landward migration needs to be fully edited for better organization. These are three separate topics but they are not clearly separated here.

We split this section into two and removed the discussion of landward migration. Previous lines discussing change were moved to the change driver section (Lines 128-150). Those discussing recovery were moved to the marsh recovery section (Lines 100-118).

Line 158: Sentence starting “Two years after..” seems extraneous and too detailed for a paper on a global mapping effort and should be removed.

We have revised the sentence slightly. However, a major portion of global salt marsh extent is within the Gulf of Mexico, and significant research has been conducted there. The majority of recovery was found within this region, and as such, we feel this is relevant to highlight the need

to study recovery at scale and potential of regional restoration efforts.

Figure 4: The error bars in the bar charts have not been defined.

We have added error bar definition to the figure caption.

Line 170: Define the LSLC acronym.

We had defined the acronym briefly and have added a more detailed explanation

Line 176: What is meant by “more impactful” here?

The term and section clarified that in both models, gain in the surrounding area was more significantly tied to loss and gain in the marsh.

Line 182: run-on sentence

The sentence was split into two sentences to resolve the run-on problem.

Line 209: Add “USA” to Boston

Added USA.

Line 217: Statement “The Clean Water Act of 1972 protects all wetlands” is not true. It only protects wetlands adjacent to navigable waterways.

We clarified this in the text. Stating on line 257 "the Clean Water Act of 1972 protects wetlands connected to navigable water in the US."

Lines 219 – 223 are hard to follow, are not logical, and should be dropped.

We removed this section as suggested.

Line 254: Define the NDC acronym.

The NDC acronym was defined on first mention at line 240

Methods

Line 262: How did you address the line striping in Landsat 7?

We added additional detail to the section. Clarifying that we filter per pixel with the fmask.

Line 265: Why is NDVI the most appropriate vegetation index? Many more have been found to be much more effective for mapping wetland vegetation, like the wide dynamic range vegetation index for example, or soil adjusted vegetation index.

We removed the most appropriate language and discussed a more comprehensive range of vegetation indices. Many indices have been found to be uniquely useful in salt marshes from SAVI, WDRVI, NDVI, and Red Edge-based indices^{3,4,5,6,8,11,13}. Ultimately all these indices utilize the relationship between red and NIR bands with various degrees of stretch added compared to NDVI.-NDVI is the most common and therefore has a certain understanding across the remote sensing and general science community that other indices do not.

We detail the link between indices and change in aboveground biomass and the utilization of NDVI vs. other possible vegetation indices¹³. We add extensive citations that demonstrate successful biomass prediction and changes in vegetation with NDVI, WDRVI, and SAVI^{3,4,5,6,11}. Ultimately, these indices are composed of red and NIR bands, making them highly correlated. The benefits of using NDVI, i.e., longest and one of the most frequently utilized indices, outweigh the potential advantages of other indices. Machine learning approaches such as kNDVI are discussed, but similar to WDRVI, address saturation, a problem that is less likely to occur in tidal marshes than forests. The anomaly analysis reduces the effect of the high tidal stage impacts by comparing pixel-wise across time to a long-baseline dataset which likely has some tidal influence in the original reference data. This analysis approach has consistently been used to understand forested tidal marshes and is thus utilized here^{14,15,16,17}.

Line 270: What is meant by minimal inclusions?

We clarify that this is referring to user's accuracy i.e. few stable areas were included in the changed areas (Line 317).

Line 277: It is unclear how a gain and a loss anomaly was defined. Is an increase in NDVI considered a gain?

This was discussed in the prior paragraph, and we added additional detail to clarify (Lines 313-318). A loss anomaly was an average NDVI reduction of ≤ -0.2 across the entire epoch. This is a conservative estimate that requires the location be changed in most of the included dates of the epoch. Gain anomalies were the same but for ≥ 0.2 change in NDVI across the epoch.

Line 288: The imagery within Google Earth Pro and other ancillary imagery types should be reported, including the spatial resolution. Authors should also report if assessment points covered areas of substantial change according to the time series maps.

We have added the region of all points and additional details about the Google Earth Pro imagery and ancillary data (Lines 335-337)

Line 303: Use of beta regressions here seem appropriate given the dependent variable is percent change

We agree and think this method is appropriate and have kept the section as written.

Line 312: The base map for Russia was in the 1980s. It is unclear how this early date influenced mapping results, though little change was reported in the 2000-2004 epoch but substantial change was reported in later epochs. Potentially there was marsh change from 1980 to 2000 that does not get reported.

The Russian baseline was 1984-2004 for the majority of the Siberian region. This was mentioned in the methodology section but has been clarified (see line 290). Therefore, in general, change in 2000-2004 only refers to a small salt marsh area in the western portion of Russia. As the beta regression analysis demonstrates, the change anomaly analysis is relatively robust to earlier mapping data. While an influence is evident, it isn't overwhelming. For example, a marsh pixel that became water during the reference period is unlikely to be included as loss due to the relatively stable low NDVI, i.e., without a transition between states in the reference to monitoring period, change is unlikely to be detected.

Line 320: Panel regressions are an appropriate method here given this is time series data for a series of watersheds.

We agree and have kept the section as written. The panel regression has been moved to the supplemental information.

Line 337: What is Hurdat2 data?

Additional description of Hurdat2 has been added to the section (Lines 397-404).

"Hurricane track and intensity data (Hurdat2) were acquired from National Weather Service and processed using the R package 'tidyverse'"

Line 351: Salt marsh carbon stocks are not considered ecosystem services, though carbon sequestration is a service

The text was changed to clarify that we are discussing the impact of change on salt marsh carbon.

Supplement:

General comment: define all acronyms in table and figure captions.

We have defined all acronyms in the supplemental information.

Table S3: The caption should be rewritten to more accurately explain the table. "Reference"

and “Salt Marsh Change” are not classes.

We have clarified the caption to “Reference category refers to the classes (gain, loss, or stable) as determined with the high-resolution accuracy assessment. Salt marsh change anomaly category classes (gain, loss, or stable) are determined by the NDVI anomaly analysis.”

What year is this salt marsh extent accuracy assessment for?

The extent accuracy was conducted using 2019 imagery but was essentially for the entirety of the analysis given the low occurrence of change areas.

Sections on China reclassification with ancillary data and Oceania mangrove encroachment analysis: These sections show the inconsistencies of just relying on a simple single vegetation index threshold for a global classification. Other indices could have been included in the algorithm to make it more robust to varying conditions like these described here.

While there are some limitations on relying on a single index. These sections discuss limitations of the underlying datasets. We have added additional analysis of our loss anomalies as they related to the GLAD 2019 dataset in the USA (Line 353-357) and how the Mcowen data overlaps with Mangroves globally (179-182). We have additionally added a discussion of recent regional analyses including in China¹ and Europe²² (Lines 169-178).

Last sentences in section on Change in the marsh and neighboring area – how can erosion increase salt marsh resilience?

In existing modeling analyses, the eroded material is then deposited onto the salt marsh surface which builds elevation. The appropriate supporting reference is below²³.

Table S15: It would make sense to report values in Teragrams here. Are these annual values?

These are total values (2000-2019) and we have converted them to Tg.

Referee #2 (Remarks to the Author):

Review of Global Hotspots of Salt Marsh Change and Carbon Emissions

I enjoyed reading Global Hotspots of Salt Marsh Change and Carbon Emissions. In my opinion this work is an important contribution to blue carbon science because most change analyses to date have focused on mangrove ecosystems which have distinct advantages from a remote sensing perspective. Although my expertise is not in remote sensing or change analysis the

methods applied appear to be robust.

Major Comments

1. The change analysis provides several important and high-impact insights. Hotspots of losses in the US Gulf Coast are a well-documented phenomenon lending confidence in the methods used here. Less well understood are the losses in Alaska and Russia, which lead the authors to identify the Arctic as a large knowledge gap (L245). The epoch-based analysis provided insights on the causes of marsh loss and recovery, such as the relationship between loss and hurricane strength followed by recovery in North America. Change analysis is the primary strength of the manuscript and was appropriately placed up front.

We agree with this statement and subsequent suggestions and have revised the manuscript to focus on the change analysis results and simplified the carbon monitoring component.

2. The consequences of area change for carbon gains and losses are also a major emphasis of the work, which in my opinion is less rigorous and detracts from a complete discussion of the area change analysis. For example, a more thorough discussion of saltmarsh loss and gain in China seems important considering that that China lost globally significant areas of saltmarsh during the time period this analysis covers. China is discussed a bit in the supplementary material with respect to classification, but there is little discussion related to the so-called “Great Seawall of China” that constitutes (I assume) most of the direct human impact on coastal marshes since the later part of the 20th century; direct land use conversion is also the dominant cause of mangrove losses. By comparison, saltmarsh losses in the US, Europe, and elsewhere have been driven by less-direct human impacts such as sediment starvation, accelerated sea level rise, and coastal erosion. I suggest the authors invest more text in this short-format article to global spatial patterns in the causes of loss and gains of salt marsh area, and less text to carbon consequences. I am sure their extensive knowledge on this subject would be enlightening.

We have moved the carbon analysis later in the paper as suggested and sought to add additional rigor both with new references^{20,21} and the use of the Coastal Carbon Database¹⁹. A detailed discussion of China highlighting recent research¹ and changes in the region (Lines 169-173). We elaborated on the mangrove discussion and moved it to the global change drivers’ section to highlight global spatial patterns (Lines 179-197).

I trust the authors on their analysis but have seen the rate marsh reclamation reported as 40,000 ha/yr from 2006-2010 (Ma et al. 2016, Science); this was an opinion piece and lacked a citation but the rate is far higher than reported here (7,602 ha over 5 years) for China in Table S1. Given some of the issues with misclassification in China discussed in the supplementary material, some discussion about how the current change analysis compares to previous regional estimates seems in order.

We have added a brief discussion of the latest change analysis of salt marsh for China¹. While our mapping approach uses only half of the extent, their analysis utilizes overall net change rates for both analyses were similarly small. Suggesting that while we assume impartial extents for most of the world, our change rates are reasonable. Several months ago, we requested the data from Chen et al.¹ to run our analysis with a more complete salt marsh extent for the region and understand the changes spatially.

As a minor note, I know the term “reclamation” is used to describe conversion of coastal wetlands to agriculture, but the term carries a negative value judgement about coastal wetland ecosystem services that is counter to that in this paper. I recommend using a different term such as “drainage”.

We agree and have changed the term to drainage (Line 13, 31, 131, 166).

Another topic that is discussed only briefly in the paper and the supplement, but could use more focus, is the encroachment of mangroves into saltmarshes.

While we agree that mangrove encroachment could use additional analysis, and we added a brief analysis of the proportion of GMW mangrove pixels within the Mcowen salt marsh extent from 1996-2016. However, our analysis gives us minimal insight into encroachment besides the potential link between gain anomalies and regions of encroachment. We agree that future global analysis of tidal wetlands should continue and further explore this phenomenon.

3. The implications of area gains and losses for carbon stocks is a reasonable goal of such a paper and an expectation of readers as a logical next step. However, in my opinion the emphasis on carbon estimates is larger than warranted by the reduction in uncertainty over previous estimates. I appreciate that the authors acknowledged uncertainty in the global/regional area of coastal marshes when calculating stock change. I agree. Given the large losses in Russia and the poor representation of Northern Russia in the global map of salt marshes used in this study (along with Canada, South America, and Africa) the lack of good maps is reason enough for caution.

We agree with this recommendation and deemphasize the work's carbon accounting component by moving it to the end and simplifying the analysis (Lines 200-226). We present several possible values for emissions given existing soil organic carbon values from Coastal Carbon Atlas¹⁹, and Soilgrid250. We show several options as none are ideal for global salt marsh mapping.

The authors used a spatial database – SoilGrids -- for soil carbon estimates. The metadata for SoilGrids states that “Tropics, wetlands, semi-arid to hyper-arid areas and mountains are still largely under-represented.”. It was not clear from the supplementary information how good the database is for tidal wetlands, but it is unlikely that SoilGrids represents the best-possible source of such data. I am curious why the authors did not choose to use the Coastal Carbon Atlas which has over 6,000 soil profiles from tidal wetlands globally. There may be good reasons such as the fact that the Atlas is biased towards the US, but it also has a lot of data from outside the US. In any, case the current paper does not seem to be using the best-available soil carbon stock data. One advantage of the Atlas is that it would allow a more precise calculation of soil carbon stocks to 1 meter as the data are disaggregated by depth. Similarly, rates of soil carbon accretion were taken from an older source (published in 2013) while more recent and extensive compilations are available in the Coastal Carbon Atlas database.

We have added a compared soil organic carbon stock values derived from Alongi 2020 literature review²⁰ and an analysis of the available emergent wetland soil samples in the Coastal Carbon Research Coordination Network ¹⁹. Finding decent agreement between the two values The CCRCN was 270.4 ± 2.8 Mg/ha¹⁹, which is slightly lower than values from Alongi 2020, (317.2 ± 19.1 Mg/ha). The Atlas samples are predominately from the US and some from Europe, but the results are comparable to the Alongi numbers. We agree there are more recent data on carbon accretion in salt marshes available via Coastal Carbon Atlas database e.g., tidal wetlands²⁵, and others for mangroves²⁴, and to include recent carbon burial we utilized the recent (2021), detailed meta-analysis for carbon burial²¹.

I suggest that the authors keep the change in carbon gain in Fig 1, delete Fig 3 (which is premature), move the carbon discussion to near the end of the paper to reduce emphasis, and dedicate space to discussions that are more squarely based on area gains and losses.

We have deleted Figure 3 and moved the carbon discussion towards the end of the paper. To group change estimates and processes together for clarity and flow.

Minor Comments

1. It appears from the supplementary information that the authors estimated belowground biomass and added it to soil carbon stocks. This is problematic because soil carbon stock estimates in these systems typically include roots already (i.e. roots are rarely separated from soil cores when quantifying stocks). Thus, there may be double counting. The error is probably relatively small compared to the soil organic carbon itself, but it should be corrected. An example of where this issue was taken into account is in Rogers et al. (2019, Nature).

Belowground biomass was removed from the carbon stock estimates. We now state “Belowground biomass was assumed to be included within soil core measurements of SOCS and

was not computed separately.” (Line 416-417). This updated assumption and new values used for SOCS and carbon burial resulted in updating estimates of SOCS loss and emissions. (Lines 199-233).

2. Many of the comparisons used to help readers understand the scale of changes were not compelling. I have some idea of the area of Hong Kong but no idea about the area of its territorial waters which could be far larger (L71). I realize that salt marshes are essentially marine grasslands, but I am unclear what to conclude from comparing carbon losses in these systems to terrestrial grasslands (L107). The comparison to Indonesian oilseed production for Chinese consumption seems a little odd.

We agree many of these comparisons offered no greater clarity and were removed, including terrestrial grassland carbon and oilseed production. We altered the Hong Kong comparison to describe the loss relative to soccer fields lost per hour and kept the comparison in the abstract but add the exact area (Lines 70 and 17, respectively).

L37. The gap was never stated. Add a sentence to L36 that clearly states the gap to be filled.

We have added that modern salt marsh change rates are the primary knowledge gap identified and filled by this work. "The latest blue carbon accounting still relies on dated estimates of salt marsh change ($1-2\% \text{ yr}^{-1}$)²² derived from limited *in situ* analyses of several estuaries and century long time period^{23,24}."

L57. I suggest "...), which is similar to the lower bound and an order of magnitude..." to provide a more balanced assessment.

This change was made with appropriate changes to reflect the updated estimate of emissions. The sentence now reads, "Our updated salt marsh carbon budget found global emissions of $16.7 (-3.2-37.4) \text{ Tg CO}_2\text{e yr}^{-1}$, which is similar to the lower bound ($20 \text{ Tg CO}_2\text{e yr}^{-1}$) and an order of magnitude reduction in the upper bound ($240 \text{ Tg CO}_2\text{e yr}^{-1}$) of previous emission estimates²⁶."

L90. The authors should be cautious about some of their interpretations. I agree that there is ice in the image of the Maryland eastern shore but am quite sure that ice scour is not an important process there as it is in the northeastern US. That area does not freeze long enough to cause scour.

We agree that this would require much additional analysis to verify, and instead, other drivers, i.e., SLR, are likely to drive the change. We have adjusted the revisions to remove this assertion and focus on other change drivers like SLR (Lines 146-154).

L112. Another good citation for the depth of disturbance is Kauffman, J.B., Heider, C., Norfolk, J. and Payton, F. (2014), Carbon stocks of intact mangroves and carbon emissions arising from

their conversion in the Dominican Republic. *Ecological Applications*, 24: 518-527. <https://doi.org/10.1890/13-0640.1>

We have added a sentence discussing depth as an unquantified uncertainty. "The use of 1 m to calculate SOCS ignores another source of significant uncertainty, which is soil depth, for example, the mean depth of cores representing deposit depth in emergent vegetation was 194.5 cm (Table S14)." We added a supplemental table including core references which represent maximum depth of the core (Table S14).

L129. The sentence "including salt marshes and other tidal marshes" raises questions about what types of tidal marshes were in or out of the current analysis. What are these other tidal marshes and were they included in the change analysis?

Salt marshes and tidal marshes are often used interchangeably, and given the Mcowen map being this study's baseline, we felt salt marsh was the most appropriate term. However, tidal marshes can include freshwater marshes and forested tidal marshes. We added other tidal marshes in our uncertainty propagation. This was clarified in Lines 438-446, moving text from the supplemental into the methods.

L157. What evidence is there that restoration and natural revegetation are the causes of the apparent recovery rate?

We conducted an accuracy assessment using the same method as change analysis accuracy. Attempting to verify that a loss occurred and a subsequent return to vegetated marsh occurred. While the accuracy assessment was limited, it demonstrated that these changes visually corresponded with loss and recovery of vegetated extent. (see Table S8.). There is a likelihood that some recovery corresponded with floating vegetation. During the accuracy assessment, some of the detected recovery locations corresponded with restoration sites.

L186. Please find a more precise term in place of "vibrancy"

Vibrancy was replaced with "NDVI gain anomalies".

L188. It is not clear how this analysis detected that a site "got wetter". Please use more precise language to help the reader understand the inference.

This section was removed. To maintain a focus on our change analysis.

Referee #3 (Remarks to the Author):

This paper presents initial results from a global scale analysis of changes as gain and loss, in saltmarsh(?) inferred from changes in a vegetation index derived from the Landsat series of satellites (2000-2019), with change identified from within a historic saltmarsh boundary from Mcowen et al. (2017) in four epochs 2000-2004, 2005-2009, 2010-2014, 2015-2019. The changes were used to estimate changes in Carbon stocks at local to regional scales, providing figures that were close to, but lower than current best case estimates. The results were validated from a global series of observation points. More detailed assessment of the potential drivers of change were conducted for the continental United States using land-cover data sets.

As a whole the paper does represent a potentially useful starting point to the growing body of work that is using global satellite archives and on-line data stores and processing environments to map and measure changes in the extent and properties of a range of environments, e.g. forests, inter-tidal areas, coastlines, mangroves, seagrass, coral reefs (Runting et al., Crowther et al., Murray et al., Galiatsatos et al., Mao et al., Lyons et al.,). There are several major concerns with the content and focus of the paper, outlined below, that need to be addressed before it is in a form suitable for publication. Once these are addressed this will be a highly significant contribution to our ability to map and monitor changes in this essential ecosystem and to provide more accurate spatial data to drive C-stock and budget assessments from local to national and global scales.

We thank Dr. Phinn for his detailed and constructive comments, we have addressed each point below.

Supporting references:

Mcowen C, Weatherdon L, Bochove J, Sullivan E, Blyth S, Zockler C, Stanwell-Smith D, Kingston N, Martin C, Spalding M, Fletcher S (2017) A global map of saltmarshes. Biodiversity Data Journal 5: e11764. <https://doi.org/10.3897/BDJ.5.e11764>

Runting, R.K., Phinn, S., Xie, Z., Venter, O. and Watson, J.E., 2020. Opportunities for big data in conservation and sustainability. Nature communications, 11(1), pp.1-4.

Hanson et al.

Crowther, T.W., Glick, H.B., Covey, K.R., Bettigole, C., Maynard, D.S., Thomas, S.M., Smith, J.R., Hintler, G., Duguid, M.C., Amatulli, G. and Tuanmu, M.N., 2015. Mapping tree density at a global scale. Nature, 525(7568), pp.201-205.

Galiatsatos, N., Donoghue, D.N., Watt, P., Bholanath, P., Pickering, J., Hansen, M.C. and

Mahmood, A.R., 2020. An assessment of global forest change datasets for national forest monitoring and reporting. *Remote Sensing*, 12(11), p.1790.

Murray, N.J., Phinn, S.R., DeWitt, M., Ferrari, R., Johnston, R., Lyons, M.B., Clinton, N., Thau, D. and Fuller, R.A., 2019. The global distribution and trajectory of tidal flats. *Nature*, 565(7738), pp.222-225.

Mao, Y., Harris, D.L., Xie, Z. and Phinn, S., 2021. Efficient measurement of large-scale decadal shoreline change with increased accuracy in tide-dominated coastal environments with Google Earth Engine. *ISPRS Journal of Photogrammetry and Remote Sensing*, 181, pp.385-399.

B. Lyons, M., M. Roelfsema, C., V. Kennedy, E., M. Kovacs, E., Borrego-Acevedo, R., Markey, K., Roe, M., M. Yuwono, D., L. Harris, D., R. Phinn, S. and Asner, G.P., 2020. Mapping the world's coral reefs using a global multiscale earth observation framework. *Remote Sensing in Ecology and Conservation*, 6(4), pp.557-568.

Specific questions to address:

A. Summary of the key results

This paper presents initial results from a global scale analysis of changes as gain and loss, in saltmarsh(?) inferred from changes in a vegetation index derived from the Landsat series of satellites (2000-2019), with change identified from within a historic saltmarsh boundary from Mcowen et al. (2017) in four epochs 2000-2004, 2005-2009, 2010-2014, 2015-2019. The changes were used to estimate changes in Carbon stocks at local to regional scales, providing figures that were close to, but lower than current best-case estimates. The results were validated from a global series of observation points. More detailed assessment of the potential drivers of change were conducted for the continental United States using detailed land cover data sets.

B. Originality and significance: if not novel, please include reference

This paper is original in terms of the problem addressed, data sets used and methods applied to estimate changes in salt-marsh. The paper represents a highly useful starting point for developing a more accurate approach to mapping and measuring changes in saltmarsh extents, from local to global scales. The derivation of the research problem cites appropriate literature, as does the development and implementation of the methods used, along with the analysis and interpretation of the results. There are some limitations with the literature covered in relation to similar locally detailed, global scale change analyses, and the satellite vegetation indices used. These limitations are outlined in the detailed comments table below.

We have sought to include additional change analysis, including recent work detailing salt marsh change in China¹, global analysis of salt marsh edge erosion²⁶. A brief vegetation indices-

related literature review was also added to the methods section. Several additional global studies you suggested have been incorporated, including B.Lyons et al. 2020²⁷.

C. Data & methodology: validity of approach, quality of data, quality of presentation

This paper

Data – appropriate data and methods were used through the paper, and their selection and limitations are clearly stated and linked to appropriate literature. More details are needed on some of the data sets used (e.g. validation of land cover changes) and selection of satellite index data sets, with details outlined in the table below.

Additional details have been added to both the validation and vegetation index use, these changes are detailed in the answers to the table recommendations.

Methodology – appropriate and logical design, supported by relevant literature in their application and assessment of limitations. Several sections, as noted in the comments table below, do require expansion and clarification of the methods used and their limitations, especially validation reference data, and in some cases should be considered for removal to simplify the message of the paper (e.g. detailed analysis within US).

We have added additional details in the text and answers to detailed comments. We conducted further analyses to fulfill some of these aspects, including comparing the loss areas in the USA to the GLAD 2019 landcover data (Lines 353-357).²⁸

<https://glad.umd.edu/dataset/global-land-cover-land-use-v1>

D. Appropriate use of statistics and treatment of uncertainties

This paper does appear to use statistics and uncertainties in the mapping, saltmarsh area change and C-stock calculations.

The statistic and treatment of uncertainty were updated to be more precise about the use of confidence interval and standard error. A sentence was added to the methods (Line 452) stating that "Uncertainty are report in parentheses are 90% confidence intervals or standard error after a \pm ."

E. Conclusions: robustness, validity, reliability

The conclusions reached in this paper are primarily supported by the data sets, methods and interpretation methods. However, the scope of the findings needs to be clarified and then modified to fit within the data sets and analysis methods used, noting the reliance on a sub-optimal (but best on offer) global salt marsh extent compilation, and that this is the first step in

developing an approach able to be used for saltmarsh change analysis. Once this is done it, then the work can be shifted to assessing saltmarsh C-dynamics.

We agree with the comment and have given our best effort to revise the paper for clarity in our approach and its limitations. Adding a first citation of the Mcowen map in line 43.

F. Suggested improvements: experiments, data for possible revision
See list of requested changes in the “Specific Comments” table below.

G. References: appropriate credit to previous work?
Yes – with some required changes, see “General Comments” above and requested changes in the “Specific Comments” table below.

H. Clarity and context: lucidity of abstract/summary, appropriateness of abstract, introduction and conclusions

The paper is constructed and written clearly, and is free from grammatical and punctuation errors. The abstract is appropriate but requires several edits, see requested changes in the “Specific Comments” table below. The introduction and conclusion also require specific changes as noted in the “Specific Comments” table below.

We have revised the paper including abstract as suggested.

15 This is very abstract for the majority of readers - and Singapore has island and mainland sectionsprovide actual area in km² ...maybe from
<https://www.worldatlas.com/features/countries-by-area.html#countriesBySize> ?

We adjust the comparison and mention the actual area of Singapore.

19-20 Suggest results stay within context of the data and methods and also recognise that multiple countries and private companies are already doing this. It would be better to place this work in context of those approaches and where they are going.

The sentences were removed and replaced with

“Our data and analysis substantially advance our understanding of salt marsh distribution and contribution to blue carbon emissions. Our monitoring method clarifies current rates of global salt marsh change and provides an important step to improving our understanding of these blue carbon ecosystems.”

21-22 The findings would also be strengthened if placed in context of other estimates of global; saltmarsh extent and C dynamics e.g. Alongi, D. (2020) Carbon Balance in Salt Marsh and Mangrove Ecosystems: A Global Synthesis. J. Mar. Sci. Eng. 2020, 8(10), 767;
<https://doi.org/10.3390/jmse8100767>

Thank you—we have added several recent references into the discussion including the Alongi 2020, Macreadie 2021, Wang et al. 2021 and other related references^{19,20,21,29}. We discuss these values in the carbon section and methodology (Lines 201-207; 410-429).

35-36 This would be significantly improved by starting with "saltmarsh extent" as that is a primary data product that can then be used and assessed, and collectively improved. e.g. see the global inter-tidal and global coral reef data sets.

We have clarified that change and global/regional change rates are what we primarily derive and something desperately needed in the blue carbon community. We state, "Here, we create the first consistent spatial and temporal estimates of contemporary salt marsh change from 2000-2019."

39-40 Can you clarify in more detail what this means, especially the 'within 1.8km of the previously...'

Clarified to "We processed all Landsat 5, 7, and 8 imagery with Google Earth Engine within 1.8 km of the known extent⁹..."

41-42 So the Mcowen et al. (2017) polygons were used as the "baseline" extent and the assumption was that if the amount of vegetation changed, the NDVI value would change, i.e. anomaly detection? Why not just state that method more clearly? It also seems subject to significant bias: - extent of Mcowen et al. polygons; - assumption that saltmarsh gain or loss is shown by change in NDVI Doesn't this only show the change in NDVI valuesbut no real link to actual saltmarsh dynamics ? Case for this still needs to be made more clearly.

The change rate bias is explored with the mapping date and the accuracy assessment. NDVI has repeatedly been linked to aboveground biomass in coastal environments. The index can predict seasonal changes to biomass long-term vegetation trends and performed well in our accuracy assessment. While we did not evaluate other indices, NDVI has a certain value as an early and widely used index with frequent used in tidal wetlands^{3,4,5,10,14,15,16,17}. We have added additional support and discussion of vegetation indices and their relation to change in biomass/extent. See below and responses to reviewer 1. We believe this response and the section in the methods make clear the connection between NDVI change and ecosystem change.

47 No need to overcomplicate things.... why not just say "mainland continental US, excluding Alaska, HI and others..."

We appreciate the suggestion but CONUS is a commonly used term for what we are describing and we would be remiss not to use it.

59-60 Keep to single area unit - not ha - use km². This sentence needs more detail - as it is not clear what it means ?

We have revised most area estimates including figures 1, 2, and 3 in the paper to use km² excluding the panel analysis which makes more sense in hectares given the scale of the change being discussed.

62 Insert - at appropriate local level of details to detect salt-marshes and their changes.

We have altered the sentence in this regard stating “at appropriate spatial and temporal resolution” (Line 63).

78 coastal erosion rates ? Can you provide references to back up this assertion e.g. in terms of global shoreline change e.g. Mao, Y., Harris, D.L., Xie, Z. and Phinn, S., 2021. Efficient measurement of large-scale decadal shoreline change with increased accuracy in tide-dominated coastal environments with Google Earth Engine. *ISPRS Journal of Photogrammetry and Remote Sensing*, 181, pp.385-399.

Erosion rates are suggested as a possible explanation in **Russia**, the reference is

Overduin, P. P. *et al.* Coastal changes in the Arctic. *Geological Society, London, Special Publications* **388**, 103-129 (2014). While analyzing the relationship of our changes to erosion rates would be interesting it is outside the scope of the current paper, we agree that using the data from Mao et al. (2021) to do so in the future would be great.

Figure 2 – 94-101 This will be easier to interpret if the size of all image maps was maximized, and if each image panel was labelled with the time period it represents.

We added the dates to each image and maximized the size by in part removing panel f. The figure has also been moved to the change driver discussion and is now Figure 4.

120 Isn't this a fundamental limitation of the approach - and constrains the scope of findings to looking at change from that data set as a baseline?

We agree this is a fundamental limitation. We present this as a fundamental limitation. However, the change rates derived should be only minorly affected by the mapping date, as demonstrated in that analysis. Especially gain in 2000-2004 and loss in 2015-2019. The two change rates frequently utilized in carbon accounting studies rely on *in situ* measurements of a limited number of estuaries. Our study's limitations are minor compared to the uncertainty of carbon accounting using dated and spatially limited change rates for global analysis, i.e., a centuries-long change rate that is unlikely to be representative of current % change—we agree as stated that our data are a step towards a more robust global change dataset. While total change is likely underestimated due to the starting extent, our rates of change are

representative of actual change rates. The significant uncertainty is the extent of the marsh, which we include in our propagation of uncertainty in the carbon analysis. Future global mapping going back to 1990 or 2000 bookended by a current map will supersede our research. Still, until that becomes a reality, change rates derived from the Mcowen dataset are needed. Previous studies have compared the Mcowen map to higher quality regional extents and found minimal impact on the change analysis²². We have added references to Laengner et al. both in our main section and methods.

144-195 It is still very unclear how the methods and data used are able to accurately map saltmarsh recovery and identify landward migration (isn't this limited by the Mcowen polygons) andvery difficult to conceptualize how change drivers can be inferred, let alone assigned with any confidence ?

The analysis was limited to the US and not limited by the Mcowen map; instead, we used the NWI salt marsh extent. Additionally, change in the surrounding area was included as a metric of this uncertainty. Migration has been removed while it is an interesting analysis, it requires additional detail and analysis.

182-195 This is extremely interesting - can we do it globally with <https://esa-worldcover.org/en> or <https://livingatlas.arcgis.com/landcover/>

This is a great point. We could look into doing this globally.

237-242 Just a comment – this is a very realistic and useful assessment

We agree.

265-266 Re.NDVI Not really - the paper by Lopes et al. 2020 only covers a small saltmarsh area in Portugal - given the many challenges and limitations of NDVI with bright and dark soils, open canopies and soil moisture, there really should be more careful consideration of a more effective index?

We removed the most appropriate language and discussed a more comprehensive range of vegetation indices. Many indices have been found to be uniquely useful in salt marshes from SAVI, WDRVI, NDVI, and Red Edge-based indices^{3,4,5,6,8,11,13}. Ultimately all these indices utilize the relationship between red and NIR bands with various degrees of stretch added compared to NDVI. We were incorrect to say NDVI is the most appropriate. NDVI is the most common and therefore has a certain understanding across the remote sensing and general science community that other indices do not.

We detail the link between indices and change in aboveground biomass and the utilization of NDVI vs. other possible vegetation indices¹³. We add extensive citations that

demonstrate successful biomass prediction and changes in vegetation with NDVI, WDRVI, and SAVI^{3,4,5,6,11}. Ultimately, these indices are composed of red and NIR bands, making them highly correlated. The benefits of using NDVI, i.e., longest and one of the most frequently utilized indices, outweigh the potential advantages of other indices. Machine learning approaches such as kNDVI are discussed, but similar to WDRVI, address saturation, a problem that is less likely to occur in tidal marshes than forests. The anomaly analysis reduces the effect of the high tidal stage impacts by comparing pixel-wise across time to a long-baseline dataset which likely has some tidal influence in the original reference data. This analysis approach has consistently been used to understand forested tidal marshes and is thus utilized here^{14,15,16,17}.

284-287 How were the "points" reference data that could be used to validate the saltmarsh presence or absence - what is their physical link to the variable mapped, and how do they match in space and time?

A stratified random points across those three classes within 1.8 km of the salt marsh extent. We clarify that each point represents a pixel which was evaluated for its general class.

301 -318 This is very confusing - and I am not sure how you would be able to explain this a group of decision makers to give them confidence in the mapping and ability to identify gains and losses accurately? Can this be improved significantly?

This quantification is one of the limitations of using the Mcowen dataset. The analysis demonstrates limited to no influence of mapping data on the gain in 2000-2004 and loss in 2015-2019; these values were used in the global quantification of yearly carbon emissions due to the limited impact of this component. While it would be great to improve this with a globally consistent map of salt marsh in 2000, that product would be complicated to produce given the limits on training data.

319-349 On reflection this section of analyses and interpretation may be better served in a separate paper as it distracts and confuses the (already complicated) global story of saltmarsh gain/loss and C budgets.

While we agree that this analysis does complicate the manuscript, it is also a unique and essential component. We have kept the analysis in and restructured the paper to discuss change and lead into the change driver's analysis. We have removed the discussion of migration and LCLU change surrounding the marsh.

References

1. Chen, G., Jin, R., Ye, Z., Li, Q., Gu, J., Luo, M., Luo, Y., Christakos, G., Morris, J., He, J. and Li, D., 2022. Spatiotemporal Mapping of Salt Marshes in the Intertidal Zone of China during 1985–2019. *Journal of Remote Sensing*, 2022.

2. Camps-Valls, G., Campos-Taberner, M., Moreno-Martínez, Á., Walther, S., Duveiller, G., Cescatti, A., Mahecha, M.D., Muñoz-Marí, J., García-Haro, F.J., Guanter, L. and Jung, M., 2021. A unified vegetation index for quantifying the terrestrial biosphere. *Science Advances*, 7(9), p.eabc7447.
3. Lopes CL, Mendes R, Caçador I, Dias JM. Assessing salt marsh extent and condition changes with 35 years of Landsat imagery: Tagus Estuary case study. *Remote Sens Environ.* 2020;**247**: 111939.
4. Sun, C., Fagherazzi, S. and Liu, Y., 2018. Classification mapping of salt marsh vegetation by flexible monthly NDVI time-series using Landsat imagery. *Estuarine, Coastal and Shelf Science*, 213, pp.61-80.
5. Doughty, C.L. and Cavanaugh, K.C., 2019. Mapping coastal wetland biomass from high resolution unmanned aerial vehicle (UAV) imagery. *Remote Sensing*, 11(5), p.540.
6. Jensen, J.R., Olson, G., Schill, S.R., Porter, D.E. and Morris, J., 2002. Remote sensing of biomass, leaf-area-index, and chlorophyll a and b content in the ACE Basin National Estuarine Research Reserve using sub-meter digital camera imagery. *Geocarto International*, 17(3), pp.27-36.
7. Lumbierres, M., Méndez, P.F., Bustamante, J., Soriguer, R. and Santamaría, L., 2017. Modeling biomass production in seasonal wetlands using MODIS NDVI land surface phenology. *Remote Sensing*, 9(4), p.392.
8. Buffington, K.J., Dugger, B.D. and Thorne, K.M., 2018. Climate-related variation in plant peak biomass and growth phenology across Pacific Northwest tidal marshes. *Estuarine, Coastal and Shelf Science*, 202, pp.212-221.
9. Myers-Smith, I.H., Kerby, J.T., Phoenix, G.K., Bjerke, J.W., Epstein, H.E., Assmann, J.J., John, C., Andreu-Hayles, L., Angers-Blondin, S., Beck, P.S. and Berner, L.T., 2020. Complexity revealed in the greening of the Arctic. *Nature Climate Change*, 10(2), pp.106-117.
10. Lagomasino, D., Fatoyinbo, T., Castañeda-Moya, E., Cook, B.D., Montesano, P.M., Neigh, C.S., Corp, L.A., Ott, L.E., Chavez, S. and Morton, D.C., 2021. Storm surge and ponding explain mangrove dieback in southwest Florida following Hurricane Irma. *Nature Communications*, 12(1), pp.1-8.
11. Byrd, K. B. et al. A remote sensing-based model of tidal marsh aboveground carbon stocks for the conterminous United States. *ISPRS Journal of Photogrammetry and Remote Sensing* 139, 255-271 (2018).
12. Camps-Valls, G., Campos-Taberner, M., Moreno-Martínez, Á., Walther, S., Duveiller, G., Cescatti, A., Mahecha, M.D., Muñoz-Marí, J., García-Haro, F.J., Guanter, L. and Jung, M., 2021. A unified vegetation index for quantifying the terrestrial biosphere. *Science Advances*, 7(9), p.eabc7447.
13. O'Donnell, J.P. and Schalles, J.F., 2016. Examination of abiotic drivers and their influence on *Spartina alterniflora* biomass over a twenty-eight year period using Landsat 5 TM satellite imagery of the Central Georgia Coast. *Remote Sensing*, 8(6), p.477.

14. Taillie, P.J., Roman-Cuesta, R., Lagomasino, D., Cifuentes-Jara, M., Fatoyinbo, T., Ott, L.E. and Poulter, B., 2020. Widespread mangrove damage resulting from the 2017 Atlantic mega hurricane season. *Environmental Research Letters*, 15(6), p.064010.
15. Lagomasino, D., Fatoyinbo, T., Lee, S., Feliciano, E., Trettin, C., Shapiro, A. and Mangora, M.M., 2019. Measuring mangrove carbon loss and gain in deltas. *Environmental Research Letters*, 14(2), p.025002.
16. Zhang, C., Durgan, S.D. and Lagomasino, D., 2019. Modeling risk of mangroves to tropical cyclones: A case study of Hurricane Irma. *Estuarine, Coastal and Shelf Science*, 224, pp.108-116.
17. Mondal, P., Dutta, T., Qadir, A. and Sharma, S., 2022. Radar and optical remote sensing for near real-time assessments of cyclone impacts on coastal ecosystems. *Remote Sensing in Ecology and Conservation*.
18. Crotty SM, Ortals C, Pettengill TM, Shi L, Olabarrieta M, Joyce MA, et al. Sea-level rise and the emergence of a keystone grazer alter the geomorphic evolution and ecology of southeast US salt marshes. *Proceedings of the National Academy of Sciences*. 2020;117: 17891-17902.
19. Coastal Carbon Research Coordination Network (CCRCN). Coastal Carbon Atlas. 2019. <https://ccrcn.shinyapps.io/CoastalCarbonAtlas>.
20. Alongi, D. (2020) Carbon Balance in Salt Marsh and Mangrove Ecosystems: A Global Synthesis. *J. Mar. Sci. Eng.* 2020, 8(10), 767; <https://doi.org/10.3390/jmse8100767>
21. Wang, F., Sanders, C.J., Santos, I.R., Tang, J., Schuerch, M., Kirwan, M.L., Kopp, R.E., Zhu, K., Li, X., Yuan, J. and Liu, W., 2021. Global blue carbon accumulation in tidal wetlands increases with climate change. *National science review*, 8(9), p.nwaa296.
22. Laengner, M.L., Siteur, K. and van der Wal, D., 2019. Trends in the seaward extent of saltmarshes across Europe from long-term satellite data. *Remote sensing*, 11(14), p.1653.
23. Mariotti, G. & Carr, J. Dual role of salt marsh retreat: Long-term loss and short-term resilience. *Water Resour. Res.* **50**, 2963-2974 (2014).
24. Breithaupt, J.L., Smoak, J.M., Bianchi, T.S., Vaughn, D.R., Sanders, C.J., Radabaugh, K.R., Osland, M.J., Feher, L.C., Lynch, J.C., Cahoon, D.R. and Anderson, G.H., 2020. Increasing rates of carbon burial in southwest Florida coastal wetlands. *Journal of Geophysical Research: Biogeosciences*, 125(2), p.e2019JG005349.
25. McTigue, N., Davis, J., Rodriguez, A.B., McKee, B., Atencio, A. and Currin, C., 2019. Sea level rise explains changing carbon accumulation rates in a salt marsh over the past two millennia. *Journal of Geophysical Research: Biogeosciences*, 124(10), pp.2945-2957.
26. Peck, E.K., Wheatcroft, R.A. and Brophy, L.S., 2020. Controls on sediment accretion and blue carbon burial in tidal saline wetlands: insights from the Oregon Coast, USA. *Journal of Geophysical Research: Biogeosciences*, 125(2).
27. B. Lyons, M., M. Roelfsema, C., V. Kennedy, E., M. Kovacs, E., Borrego-Acevedo, R., Markey, K., Roe, M., M. Yuwono, D., L. Harris, D., R. Phinn, S. and Asner, G.P., 2020.

Mapping the world's coral reefs using a global multiscale earth observation framework. *Remote Sensing in Ecology and Conservation*, 6(4), pp.557-568.

28. Hansen, M.C., Potapov, P.V., Pickens, A., Tyukavina, A., Serna, A.H., Zalles, V., Turubanova, S., Kommareddy, I., Stehman, S.V., Song, X. and Kommareddy, A., 2021. Global land use extent and dispersion within natural land cover using Landsat data. *Environmental Research Letters*.
29. Macreadie, P.I., Costa, M.D., Atwood, T.B., Friess, D.A., Kelleway, J.J., Kennedy, H., Lovelock, C.E., Serrano, O. and Duarte, C.M., 2021. Blue carbon as a natural climate solution. *Nature Reviews Earth & Environment*, 2(12), pp.826-839.

Reviewer Reports on the First Revision:

Referees' comments:

Referee #1 (Remarks to the Author):

I appreciate the effort the authors put into revising the manuscript, including testing a machine learning kernel and considering other vegetation indices beside NDVI. I also appreciate the effort to clarify error terms and include error bars on tables and figures. The manuscript is now better organized.

The manuscript is still a bit lengthy and includes some extraneous information that does not fit in well in a paper on global tidal wetland mapping. There are several editorial and grammatical errors. Verb tense is not consistent. There are some broad statements on the causes of marsh loss that are not well substantiated, that need to be revised or deleted. The manuscript should be reviewed and corrected by an experienced or professional editor to make it more readable and remove irrelevant information.

Specific comments are provided below.

Line 35: Define blue carbon

Line 53: At least a sentence is needed to describe this referenced study on mangrove losses

Line 66: Captions for figures a,b, and c are missing.

Line 102: What evidence is there that recovery occurred after storm events? Add this information to this section.

Line 108: Some of this information is repeated from above.

Line 115: Include information on the area of wetlands restored through the Coastal Master Plan

Line 116: Drop the sentence starting "These projects provide.."

Line 125: Generally how did you ensure that marsh loss detected was not due to a difference in tide height between images, where higher tides could indicate marsh loss on the edges of marsh?

Line 142: Sentence ending on this line needs a reference

Line 144: Be careful in assigning cause of LSLC here. Better to say higher LSLC was related to reduced gains

Line 145: This paragraph is an example of broad statements that are not well supported, and needs substantial editing. For example, on line 147, change "such as" to "characteristic of".

Line 148: How much SLR has occurred in this section of Maryland's Eastern Shore?

Line 151: Define LSLC earlier in the manuscript

Line 160: Change "g." to "f."

Line 171: Provide more information on what is meant by "half the starting salt marsh extent"

Line 178: This paragraph would be better placed in a new section on error and need for further analysis

Line 198: It is not clear how gains in soil carbon stocks were calculated, since this typically occurs over a long time period. Provide additional clarification.

Line 218. This paragraph needs to be shortened, and could go in the new section on error. Delete "of the historic extent."

Line 268: Change to "Section 4040 of the U.S. Clean Water Act of 1972"

Line 273: What is the global tidal wetland change map? Provide more information.

Line 292: Provide information about the different epochs analyzed at the beginning of this section.

Line 310: Spell out all acronyms. This section can be greatly reduced to say that a few methods were considered but the NDVI threshold method was selected because....

Line 349: Provide the resolutions of the imagery sources listed here.

Line 358: By 0.2, do you mean the absolute value of 0.2, since you were considering gains and losses?

Line 370: Mapping year – is this the starting mapping year? Clarify in the text.

Line 391: Change this section heading to something like "Drivers of marsh change."
Line 425: Specify that you are referring to soil carbon loss down to 30cm, which is shown in the supplement.
Line 428. The Byrd et al. 2018 paper has a correction. Byrd et. al. 2020 <https://doi.org/10.1016/j.isprsjprs.2020.05.005>. Be sure to cite the correction and use the updated values.
Line 442: The comparison with the SoilGrid data should be greatly shortened or deleted.

Supplement

Tables S3-S5. Were the number of accuracy assessment points the same for each time period, as shown in the caption? Check captions for typos – the phrase "was the class" is repeated.

Table S7. Caption also needs to be edited.

Mapping year uncertainty section: Starting off with NWI information does not seem relevant here. Edit this section for clarity. In terms of Mapping year, do the authors mean starting mapping year?

Change in the marsh and neighboring area: This section needs substantial editing. Check the accuracy of mapped woody and herbaceous wetlands in the National Landcover Database. It is not high. As a result I would be cautious about the broad interpretations made from the change data. Drop the sentence "Salt marsh edge erosion has been observed to increase salt marsh platform resilience." While eroded soils could redeposit onto marshes, broadly speaking, erosion is a major cause of salt marsh loss. Given the error associated with the NLCD change product, I would consider dropping this section.

Referee #2 (Remarks to the Author):

I thank the authors for carefully considering my comments on their manuscript. They made extensive edits to their analyses and discussion that substantially improve the paper. am quite satisfied but have a few additional comments related to my primary point about the work being primarily focused on area change analysis.

L13. Suggest editing to: "Here we conduct a global and systematic change analysis of Landsat satellite imagery from the years 2000 to 2019 to quantify loss, gain, then estimate recovery of salt marsh ecosystems and the impact of these changes on blue carbon stocks". Adding "then estimate" signals to the reader that the primary purpose of the work was not to quantify carbon stocks.

L54. Please delete the sentence "Our updated salt marsh carbon budget found global emissions of 16.3 (0.42-33.4) Tg CO₂e yr⁻¹, which is similar to the lower bound (20 Tg CO₂e yr⁻¹) and an order of magnitude reduction in the upper bound (240 Tg CO₂e yr⁻¹) of previous emission estimates." It is out of place in this paragraph on change analysis. It is better to leave any discussion of the carbon implications to the later section that begins on L200.

Figure 2. Panel b should be taken out and made into a separate figure that appears with the carbon discussion on L200. It is now out of place with the reorganization of the paper.

L204. There seems to be a decimal point missing in "(-014)".

L204. I do not understand the sentence "the net change in carbon burial does represent the potential that current change already represents a net sink of carbon when considering only burial." I cannot tell whether there are missing words, confusion from two uses of "represent", or

problems conveying the essential thoughts involved.

-- Pat Megonigal

Referee #3 (Remarks to the Author):

The authors have completed thorough and extensive revisions to address in full the majority of critical comments and requested changes from all three reviewers.

The authors should be commended on a very thorough set of revisions, and well written response to reviewer comments.

My only sticking point, as this is a global journal , and we are trying to be non-US centric, the US way is not the only way... :-) is the following must be addressed:

47 No need to overcomplicate things.... why not just say "mainland continental US, excluding Alaska, HI and others..."

Once that is done, the paper is in publishable form.

Referees' comments:

Referee #1 (Remarks to the Author):

I appreciate the effort the authors put into revising the manuscript, including testing a machine learning kernel and considering other vegetation indices beside NDVI. I also appreciate the effort to clarify error terms and include error bars on tables and figures. The manuscript is now better organized.

The manuscript is still a bit lengthy and includes some extraneous information that does not fit in well in a paper on global tidal wetland mapping. There are several editorial and grammatical errors. Verb tense is not consistent. There are some broad statements on the causes of marsh loss that are not well substantiated, that need to be revised or deleted. The manuscript should be reviewed and corrected by an experienced or professional editor to make it more readable and remove irrelevant information.

We thank the reviewer for the continued quality guidance and have sought to address the remaining issues in this final revision. We have gone through the manuscript and worked to reduce the length, including the methods section. We have updated the verb tense to be consistent.

Specific comments are provided below.

Line 35: Define blue carbon

We have added a brief definition of blue carbon on line 36-37 i.e., “carbon-dense coastal wetland ecosystems”

Line 53: At least a sentence is needed to describe this referenced study on mangrove losses

Added a sentence on Line 57, describing the study.

Line 66: Captions for figures a,b, and c are missing.

Added captions for each inset of Figure 1.

Line 102: What evidence is there that recovery occurred after storm events? Add this information to this section.

We have added additional support and moved relevant from later in the paragraph and the previous paragraph. Now stating on line 100, “Globally, 4.7% of all salt marsh losses had

recovered by 2019, with most of the recovery occurring in areas lost between 2005-2009. These 2005-2009 losses coincide with extreme weather events such as hurricanes Rita, Wilma, and Katrina in 2005 which greatly affected the US Gulf Coast and resulted in a conversion of 562 km² of land to water in Louisiana²⁶. The 16.5% recovery rate for losses occurring from 2005-2009 in the Gulf of Mexico region provides further evidence that storm events had a higher recovery rate than other loss drivers.”

Line 108: Some of this information is repeated from above.

The sentence was removed.

Line 115: Include information on the area of wetlands restored through the Coastal Master Plan

Added a sentence, Line 117, addressing area of wetland restoration “The area of direct restoration is unclear given the potential indirect benefit of oyster reef restoration and sediment pipelines.”

Line 116: Drop the sentence starting “These projects provide..”

We deleted the suggested sentence.

Line 125: Generally how did you ensure that marsh loss detected was not due to a difference in tide height between images, where higher tides could indicate marsh loss on the edges of marsh?

Change was not dependent on any single image to image comparison instead each quality pixel is compared to our baseline. This means that all quality pixels in an epoch were compared to the baseline. Tidal influence is likely to be minimal unless there is a change in tidal inundation between the baseline and epoch i.e., a consistent long-term change. That would likely be indicative of a change in the marsh anyway. See line 307 for a sentence detailing this.

Line 142: Sentence ending on this line needs a reference

Removed several of the processes and referenced Figure 3 to identify an area of accretionary coast identified as a marsh gain.

Line 144: Be careful in assigning cause of LSLC here. Better to say higher LSLC was related to reduced gains

The beginning of the sentence was removed and instead LSLC relation to gains was highlighted. See line 141, "Higher LSLC was significantly related to reduced salt marsh gains."

Line 145: This paragraph is an example of broad statements that are not well supported, and needs substantial editing. For example, on line 147, change "such as" to "characteristic of".

Made this change and added a reference to the closest tidal gauge which demonstrates the rapid rate of SLR in the region. Line 147, "... supported by the nearby Ocean City, USA tidal gauge with a long term SLR trend of $6.05 \pm 0.73 \text{ mm yr}^{-1}$;²⁹ "

Line 148: How much SLR has occurred in this section of Maryland's Eastern Shore?

The closest tidal gauge reports 6.05 mm yr^{-1} of SLR, we have added this to the text (line 147) and think it provides support for the idea of a SLR driven loss in the region.

Line 151: Define LSLC earlier in the manuscript

This description was moved to the first mention of the term on line 54.

Line 160: Change "g." to "f."

This change was made.

Line 171: Provide more information on what is meant by "half the starting salt marsh extent"

The new Chinese salt marsh extent dataset had $\sim 1000 \text{ km}^2$ and our analysis baseline data only had $\sim 500 \text{ km}^2$. While this excludes some areas from our change analysis these are included in our upper estimate of total salt marsh extent. Line 167, the sentence was revised "Despite our analysis including only approximately half the salt marsh extent in China for 2000 (514 km^2 compared to 1176 km^2), we found a similar small net loss rate of 0.006 (-0.45 to 0.47) $\% \text{ yr}^{-1}$ and Chen et al. found a loss rate of $0.0009 \text{ \% yr}^{-1}$.³¹"

Line 178: This paragraph would be better placed in a new section on error and need for further analysis

Added a section titled uncertainty and future analysis.

Line 198: It is not clear how gains in soil carbon stocks were calculated, since this typically occurs over a long time period. Provide additional clarification.

Clarified this stating on line 227, “Conversely, gain was estimated with carbon burial rates, which, increased linearly due to the process being cumulative” and an additional sentence on line 435 was added to the methods to clarify this “Carbon burial rates were used to calculate carbon increases from gains in salt marsh extent.” We think this is a reasonable method for calculating the effect of gains due to their limited but continuous contribution to carbon stocks.

Line 218. This paragraph needs to be shortened, and could go in the new section on error. Delete “of the historic extent.”

The section was shortened and moved to the new section on error. The “of the historic extent.” Statement was deleted.

Line 268: Change to “Section 4040 of the U.S. Clean Water Act of 1972”

This change was made on Line 276.

Line 273: What is the global tidal wetland change map? Provide more information.

Added additional details and moved the paragraph to uncertainty and future analysis section. Line 203 stating, “In comparison, using the change rates derived from the recent global tidal wetland map, a map of tidal flats, tidal marsh and mangroves³⁹, results in a much lower estimate of 5.19 (0.06-10.79) Tg CO₂e yr⁻¹.”

Line 292: Provide information about the different epochs analyzed at the beginning of this section.

Added a brief sentence discussing epochs on Line 306, stating, “These 5-year epochs from 2000-2004, 2005-2009, 2010-2014, and 2015-2019 allowed for a per pixel analysis of change to minimize the effect of tidal stage. “

Line 310: Spell out all acronyms. This section can be greatly reduced to say that a few methods were considered but the NDVI threshold method was selected because.....

We significantly reduced the first paragraph of the section and believe we now have defined all acronyms on first mention.

Line 349: Provide the resolutions of the imagery sources listed here.

Added a sentence stating on Line 349, “The spatial resolution of these images varies but very high resolution (<3m) imagery was used in the majority of our land cover verification. In limited instances, for example, Alaska 30 m imagery was also utilized.”

Line 358: By 0.2, do you mean the absolute value of 0.2, since you were considering gains and losses?

Absolute was added to the sentence.

Line 370: Mapping year – is this the starting mapping year? Clarify in the text.

On line 372, clarified this to “The mapping year, the year imagery was acquired...”

Line 391: Change this section heading to something like “Drivers of marsh change.”

The section title was changed to Drivers of salt marsh change.

Line 425: Specify that you are referring to soil carbon loss down to 30cm, which is shown in the supplement.

SOCS loss is estimated for both 30 cm and 100 cm and this is clear in the text. On line 428, we clarified this point at the suggested line. “These losses were calculated for both 30 cm and 100 cm to cover a range of loss estimates. “

Line 428. The Byrd et al. 2018 paper has a correction. Byrd et. al.

2020 <https://doi.org/10.1016/j.isprsjprs.2020.05.005>. Be sure to cite the correction and use the updated values.

Values were used from the updated dataset and the citation was added to reflect this.

Line 442: The comparison with the SoilGrid data should be greatly shortened or deleted.

Given our refocusing away from SoilGrid we have removed the comparison.

Supplement

Tables S3-S5. Were the number of accuracy assessment points the same for each time period, as shown in the caption? Check captions for typos – the phrase “was the class” is repeated.

We removed the second instance of “was the class” and checked the captions for typos make several additional edits for clarity.

Table S7. Caption also needs to be edited.

The caption was edited for clarity.

Mapping year uncertainty section: Starting off with NWI information does not seem relevant here. Edit this section for clarity. In terms of Mapping year, do the authors mean starting mapping year?

We added an introductory sentence describing why we are talking about the NWI and using it in this case. Mapping year refers to the year aerial data was acquired to map an area. We also clarified that we are referring to the baseline mapping year here.

Change in the marsh and neighboring area: This section needs substantial editing. Check the accuracy of mapped woody and herbaceous wetlands in the National Landcover Database. It is not high. As a result I would be cautious about the broad interpretations made from the change data. Drop the sentence “Salt marsh edge erosion has been observed to increase salt marsh platform resilience.” While eroded soils could redeposit onto marshes, broadly speaking, erosion is a major cause of salt marsh loss. Given the error associated with the NLCD change product, I would consider dropping this section.

We agree and have deleted the section from the supplementary information. While we think it could be an informative analysis it requires more analysis and background and does not add much to the paper in its current form.

Referee #2 (Remarks to the Author):

I thank the authors for carefully considering my comments on their manuscript. They made extensive edits to their analyses and discussion that substantially improve the paper. am quite satisfied but have a few additional comments related to my primary point about the work being primarily focused on area change analysis.

We appreciate your expertise and essential feedback to the manuscript. We have addressed all remaining comments and made requested changes.

L13. Suggest editing to: “Here we conduct a global and systematic change analysis of Landsat satellite imagery from the years 2000 to 2019 to quantify loss, gain, then estimate recovery of salt marsh ecosystems and the impact of these changes on blue carbon stocks”. Adding “then estimate” signals to the reader that the primary purpose of the work was not to quantify carbon stocks.

We revised the sentence in the suggested manner.

L54. Please delete the sentence “Our updated salt marsh carbon budget found global emissions of 16.3 (0.42-33.4) Tg CO₂e yr⁻¹, which is similar to the lower bound (20 Tg CO₂e yr⁻¹) and an order of magnitude reduction in the upper bound (240 Tg CO₂e yr⁻¹) of previous emission estimates.” It is out of place in this paragraph on change analysis. It is better to leave any discussion of the carbon implications to the later section that begins on L200.

The sentence was deleted.

Figure 2. Panel b should be taken out and made into a separate figure that appears with the carbon discussion on L200. It is now out of place with the reorganization of the paper.

Figure 2 was split into table 1 and Figure 4. The figure was moved to the carbon monitoring section.

L204. There seems to be a decimal point missing in “(-014)”.

The decimal point was added.

L204. I do not understand the sentence “the net change in carbon burial does represent the potential that current change already represents a net sink of carbon when considering only burial.” I cannot tell whether there are missing words, confusion from two uses of “represent”, or problems conveying the essential thoughts involved.

We agree, the sentence was revised for clarity. “The lower bound of net carbon burial represents the potential that current change already represents a net sink when considering only carbon burial.”

-- Pat Megonigal

Referee #3 (Remarks to the Author):

The authors have completed thorough and extensive revisions to address in full the majority of critical comments and requested changes from all three reviewers.

We thank you for your essential feedback and comments. We have addressed your final critique and made appropriate changes to the manuscript.

The authors should be commended on a very thorough set of revisions, and well written response to reviewer comments.

My only sticking point, as this is a global journal , and we are trying to be non-US centric, the US way is not the only way... :-) is the following must be addressed:

47 No need to overcomplicate things.... why not just say "mainland continental US, excluding Alaska, HI and others..."

We agree with the sentiment and have made the suggested change. We continue to use CONUS as an abbreviation to describe the area of that portion of our study.

Once that is done, the paper is in publishable form.